



# Evapotranspiration over agroforestry sites in Germany

Christian Markwitz[1], Alexander Knohl[1,2], and Lukas Siebicke[1]

[1]University of Goettingen, Bioclimatology, Faculty of Forest Sciences and Forest Ecology, Germany
[2]University of Goettingen, Centre of Biodiversity and Sustainable Land Use (CBL), Germany

**Correspondence:** Christian Markwitz (christian.markwitz@forst.uni-goettingen.de)

**Abstract.** In past years the interest in growing crops and trees for bioenergy production increased. One agricultural practice is the mixed cultivation of fast growing trees and annual crops or perennial grass-lands on the same piece of land, referred to as one type of agroforestry. The inclusion of tree strips into the agricultural landscape has been shown - on the one hand - to lead to reduced wind speeds and higher carbon sequestration above-ground and in the soil. On the other hand, concerns have been

risen about increased water losses to the atmosphere via evapotranspiration (ET). Therefore it was our main objective to proof if agroforestry systems have higher ET compared to monoculture systems.

We followed a replicated measurement design to investigate the impact of agroforestry (AF) on ET under consideration of different ambient conditions. We measured actual ET at five agroforestry sites in direct comparison to five monoculture (MC) sites in Northern Germany in 2016 and 2017. We used an eddy covariance energy balance set-up (ECEB) and a low-cost eddy

covariance set-up (EC-LC) to measure actual evapotranspiration over each agroforestry and each monoculture system. We conducted direct eddy covariance (EC) measurement campaigns of approximately four weeks duration for method validation.

Results from the short-term measurement campaigns showed a high agreement between ET from EC-LC and EC, indicated by slopes of a linear regression analysis between 0.86 and 1.3 ($R^2$ between 0.7 and 0.94) across sites. Root mean square errors of LE from EC-LC vs. EC were half as small as from ECEB vs. EC, indicating a superior performance of EC-LC compared to

ECEB.

The overall effect of agroforestry on system-scale ET for the two years was small compared to the monoculture systems. Differences between annual ET over AF and MC from the two years and both measuring set-ups were not significant (p = 0.3557). We interpret this as an effect of compensating small-scale differences in ET when ET is measured on system-scale. A reduction of ET is expected to be strongest next to the tree strips due to reduced wind speed and limited incident radiation

relative to an open field. Whereas in between the tree strips ET is expected to increase due to higher incident radiation. Most likely differences in ET rates next to and in between the tree strips are of the same order of magnitude and compensate each other on system scale. In contrast, we found a strong dependency of ET on the local climate, characterized by the evapotranspiration index ($\sum$ET/precipitation). We observed significant (p = 0.0007098) higher mean evapotranspiration indices across sites for a drier than normal year (2016) compared to a wet year (2017) independent of the land-use or method.

We conclude that agroforestry has not resulted in an increased water loss to the atmosphere indicating that agroforestry in Germany can be a land-use alternative to conventional agriculture.





# 1 Introduction

In past years the interest in growing crops and trees for the production of bioenergy has increased, especially in the scope of climate change mitigation and carbon sequestration (Fischer et al., 2013; Zenone et al., 2015). One method of efficient biomass

production is the cultivation of short rotation coppice (SRC), referred to as "any high-yielding woody species managed in a coppice system" (Aylott et al., 2008). Typically, fast growing tree species, such as poplar or willow are used for SRC plantations. The trees are commonly harvested after a three to five year rotation period and used for energy and heat production (Aylott et al., 2008). SRC plantations are comparable to monoculture systems with a single tree species grown.

The cultivation of fast growing trees with annual crops or perennial grass-lands on the same piece of land is an example

of agroforestry (AF) (Morhart et al., 2014; Smith et al., 2013) and has numerous environmental benefits (Quinkenstein et al., 2009). De Stefano and Jacobson (2018) found that the inclusion of fast growing trees arranged into tree strips (short rotation alley-cropping) leads to a higher carbon sequestration above-ground and in the soil relative to monoculture systems. The additional biomass input from litter, dead wood and roots led to increased soil fertility (e.g. (Beuschel et al., 2018; Quinkenstein et al., 2009; Tsonkova et al., 2012)). Böhm et al. (2014) and Kanzler et al. (2018) reported reduced wind velocity leewards of

the tree strips when oriented perpendicular to the prevailing wind direction. In addition, Cleugh (1998) and Quinkenstein et al. (2009) found that tree strips reduce incident solar radiation, leading to reduced air temperature (McNaughton, 1988). Effects of tree strips on microclimate are mostly attributed to a region next to the tree strips with the extent depending on tree strip properties, such as the space between the tree strips, their orientation relative to the prevailing wind direction, their density, height and width (Quinkenstein et al., 2009).

Evapotranspiration (ET) in AF is strongly affected by the tree strip properties and is the combined process of 1) evaporation from the soil and open water from leaf surfaces and 2) leaf transpiration (Katul et al., 2012). ET within AF is reduced on the downwind side of the tree strips (Cleugh, 1998; Davis and Norman, 1988; Kanzler et al., 2018; Quinkenstein et al., 2009; Tsonkova et al., 2012). Davis and Norman (1988) explained the reduction in ET by the protection of adjacent crops from dry air advection. The reduced dry air advection leads to a decreased vapour pressure deficit (VPD), lowering ET (Kanzler et al.,

2018). The reduction in ET in the vicinity of the tree strips leads to an increased soil water content downwind, with the potential for enhancing yield production (Kanzler et al., 2018; Swieter et al., 2019).

Currently, little is known about system-scale water use of heterogeneously shaped alley-cropping systems in Germany. The majority of previous studies focused on the water use of short rotation coppices, but not AF systems (Bloemen et al., 2016; Fischer et al., 2013; Schmidt-Walter et al., 2014; Fischer et al., 2018). Fischer et al. (2013) and Zenone et al. (2015) observed

a lower annual sum of evapotranspiration over a poplar SRC in the Czech Republic and in Belgium compared to the annual sum of evapotranspiration over a reference grassland. This is contradictory to the assumption that SRC plantations are strong water consumers. For agroforestry systems we formulated the same hypothesis such as system-scale evapotranspiration over agroforestry systems is higher compared to monoculture agriculture.





However, the effect of agroforestry on system-scale evapotranspiration is site specific and depend on the local climate, the soil type, the water availability as well as the agroforestry design. Therefore, repeated measurements at different sites are essential for studies on the effects of agroforestry on evapotranspiration. Nevertheless, this requires methods of low maintenance with low power consumption, and moderate cost.

The most common approach for evapotranspiration measurements at ecosystem scale is the eddy covariance (EC) method (Baldocchi, 2003, 2014). EC provides a tool for real time flux measurements on a time scale of 30 minutes. The complexity and cost of traditional EC systems, however, usually limits the required replication of measurement units (Hill et al., 2017). An alternative method with lower costs is the eddy covariance energy balance method (ECEB) (Amiro, 2009). The latent heat flux (LE) is calculated as the residual of the energy balance components, i.e., the net radiation, the ground heat flux, the sensible

heat flux and various storage terms. The accuracy of the ECEB method is limited by the ability to close the surface energy balance (Foken, 2008a; Foken et al., 2010; Gao et al., 2017), because any non-closure would affect the flux estimates, typically leading to an overestimation of latent heat fluxes. Therefore, we need to assess to what extent the energy balance is closed at the given sites. Another alternative method for measurements of evapotranspiration is the use of slower but cheaper humidity sensors resulting in a low-cost eddy covariance set-up (EC-LC) (Markwitz and Siebicke, 2019). The measurement principle

follows the concept of the eddy covariance method, however, the fast response gas analyser is replaced by a slow response thermohygrometer. The slow response time of the humidity sensor limits the sampling of turbulent eddies across the whole energy spectrum, which we address by appropriate high-frequency corrections during preprocessing. For latent heat fluxes obtained by EC-LC the non-closure of the energy balance causes a flux underestimation as observed for traditional EC set-ups. Any potential non-closure we then address by direct measurements of the latent heat flux to estimate the energy balance

non-closure and partition the residual energy to the sensible and latent heat flux.

The main hypothesis of the current work was that AF systems have higher water losses to the atmosphere via ET, compared to monoculture agriculture. In order to test the hypothesis the main objectives of the study are (1) to evaluate the eddy covariance energy balance (ECEB) and low-cost eddy covariance (EC-LC) method against direct eddy covariance (EC) measurements, and (2) to measure actual evapotranspiration of five AF systems in Germany and compare those to five monoculture systems

in close vicinity to the AF systems using the two different approaches.

## 2   Materials and methods

### 2.1   Site description

This study was carried out as part of the project 'Sustainable Intensification of Agriculture through Agroforestry' (SIGNAL, http://signal.uni-goettingen.de/, last access: 19 January 2020), investigating the sustainability of AF systems in Germany. We

performed measurements at five sites across Northern Germany (Fig. 1 left). Each site consisted of one AF plot and one monoculture (MC) plot (Figs. 1 b, d, f, h and j for an aerial photograph of the Dornburg, Forst, Mariensee, Reiffenhausen and Wendhausen site with the AF and the MC selected). The AF plots are of a short rotation alley cropping type, with fast growing trees interleaved by either crops or perennial grasslands (Figs. 1 a, c and i for cropland AF systems and Figs. 1 e and





g for grassland AF systems). The crops and grasses at the monoculture plots undergo the same tillage and fertilization as the crops and grasses cultivated between the tree strips. The MC plot serves as a reference to the AF plot. Table 1 specifies the site locations and the AF geometry.

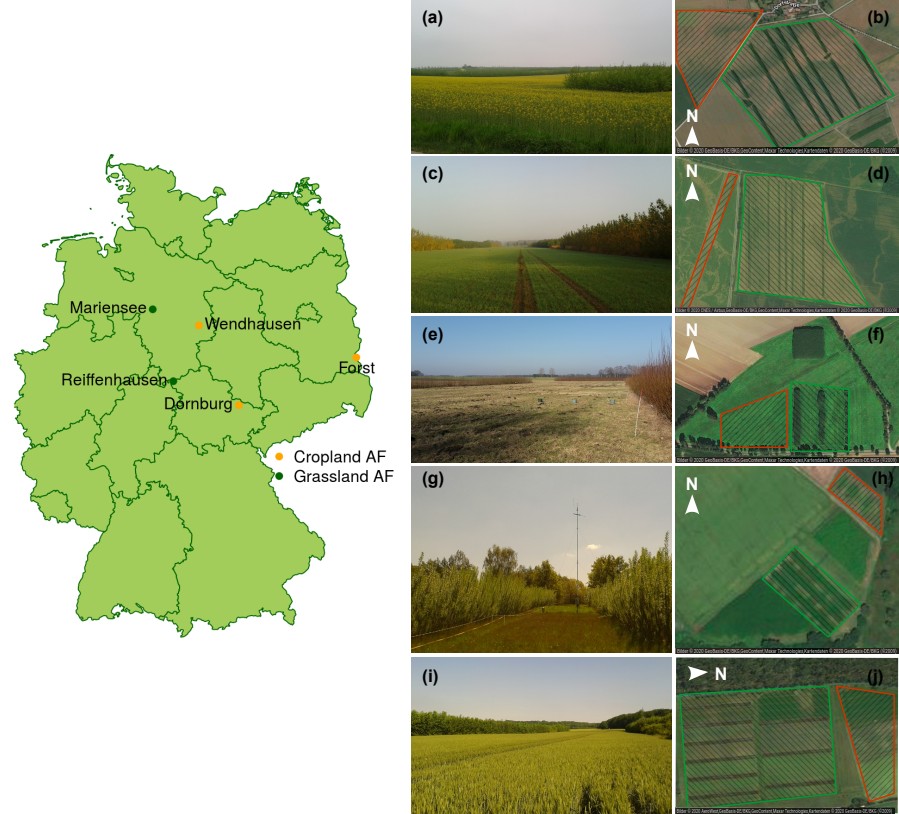

**Figure 1. Left:** map of SIGNAL sites, with the respective agroforestry type; **Right:** image and aerial photograph of the AF systems of Dornburg, (a) and (b), Forst, (c) and (d), Mariensee, (e) and (f), Reiffenhausen, (g) and (h), and Wendhausen, (i) and (j). Green hatched areas in the aerial photograph correspond to the area of the AF system and red hatched areas correspond to the area of the MC system. Site images are own photographs and aerial photographs originate from Google maps/ Google earth ©Google 2020.

## 2.2 Measurements

5  Measurements of meteorological and micrometeorological variables were performed since March 2016. At each AF plot we installed an eddy covariance mast with a height of 10 m and at each MC plot an eddy covariance mast with a height of 3.5 m. Each mast was equipped with the same meteorological and micrometeorological instrumentation. The standard set-up consisted of instruments measuring wind speed, wind direction, sensible heat flux, net radiation, global radiation, air temperature, relative humidity, precipitation and ground heat flux. An overview of the installed instruments and the respective variables used for the
10  presented set-ups is given in Table 2.



**Table 1.** Site locations and AF geometry.

| Site | Coordinates | No. of tree strips | Distance between tree strips (m) | Orientation of tree strips | Tree height (m) | Agroforestry type | System size (ha) | Relative tree cover (%) |
|------|-------------|--------------------|----------------------------------|----------------------------|-----------------|-------------------|------------------|-------------------------|
| Reiffenhausen | 51°24'N 9°59'E | 3 | 9 | NW-SE | 4.73±0.32 (n=69) Malec (2017) | Willow-grassland | 1.9 | 72 |
| Mariensee | 52°34'N 9°28'E | 3 | 48 | N-S | 4.01±0.33 (n=96) Swieter and Langhof (2017) | Willow-grassland | 7 | 6 |
| Wendhausen | 52°20'N 10°38'E | 6 | 24, 48, 96 | N-S | 6.21±0.4 (n=114) Swieter and Langhof (2017) | Poplar-cropland | 18 | 11.52 |
| Forst | 51°47'N 14°38'E | 7 | 24, 48, 96 | N-S | 6.5±1.8 (n=161) Seserman (2017) | Poplar-cropland | 39.1 | 12 |
| Dornburg | 51°00'N 11°38'E | 7 | 48, 96, 125 | NW-SE | 6.4±0.64 (n=160) Rudolf (2017) | Poplar-cropland | 51 | 8 |

Gaps in precipitation measurements at all sites were filled by precipitation data collected at nearby weather stations operated by the German weather service (DWD). We used the R-package rdwd (Boessenkool, 2019) for data download from the ftp server maintained by the DWD. We replaced gaps in precipitation measurements with DWD data if more than 25 % of precipitation data per day were missing. We used precipitation data from the weather stations Erfurt-Weimar airport, Cottbus, Hannover-Herrenhausen and Braunschweig to fill data gaps in precipitation at Dornburg, Forst, Mariensee and Wendhausen, respectively. In Reiffenhausen we used the precipitation records of a station placed at the same site and operated by the soil hydrology group at the University of Göttingen. As the precipitation transmitter was placed inside or next to the tree strips at the majority of the AF plots, the measurements were affected by interception and were lower than at the MC plot. Therefore, we used the precipitation measurements from the MC plot to compute ratios of annually summed actual and potential ET to precipitation at both AF and MC plots.

In the following sections we briefly describe the concepts of the used set-ups, eddy covariance (EC), eddy covariance energy-balance (ECEB) and low-cost eddy covariance (EC-LC). Throughout the manuscript we use the respective abbreviations.

### 2.2.1 Eddy Covariance (EC)

Sensible heat and momentum fluxes have been measured continuously with ultrasonic anemometers since 2016. The water vapour and $CO_2$ mole fraction were measured during field campaigns during the vegetation periods of 2016 and 2017. During the time of operation the standard set-up was extended by an enclosed-path infrared gas analyser (LI-7200 , LI-COR Inc., Lincoln, Nebraska, USA). In 2016, the campaigns were conducted separately at AF and MC, whilst in 2017 both plots were sampled simultaneously. All measuring periods are summarized in Table A1. Data processing and the analysis procedure is described in more detail in Markwitz and Siebicke (2019).





**Table 2.** Instrumentation for flux and meteorological measurements used at all five AF and MC plots. Set-up corresponds to eddy covariance, EC, low-cost eddy covariance, EC-LC, and eddy covariance energy balance, ECEB.

| Variable | Height (m) | Instrument | Company | Set-up |
|---|---|---|---|---|
| 3D wind components, u, v, w ($\mathrm{m\,s^{-1}}$) | 3.5,10 | uSONIC-3 Omni | METEK GmbH | EC, ECEB, EC-LC |
| ultrasonic temperature, $T_s$ (°C), wind speed ($\mathrm{m\,s^{-1}}$), -direction (°) | | | Elmshorn, Germany | |
| Net radiation, $R_N$ ($\mathrm{W\,m^{-2}}$) | 3, 9.5 | NR-Lite2 Net Radiometer | Kipp&Zonen | ECEB |
| | | | Delft, The Netherlands | |
| Global radiation, $R_G$ ($\mathrm{W\,m^{-2}}$) | 3, 9.5 | CMP3 Pyranometer | Kipp&Zonen | |
| | | | Delft, The Netherlands | |
| Relative humidity, RH (%), air temperature, T (°C) | 2 | Hygro-Thermo Transmitter-compact | Thies Clima | EC, ECEB |
| | | (Model 1.1005.54.160) | Göttingen, Germany | |
| RH, T, Atmospheric pressure, ppp (Pa) | 0.5, 3/9.5 | BME280 | BOSCH, Germany | EC-LC |
| Precipitation (mm) | 1 | Precipitation Transmitter | Thies Clima | |
| | | (Model 5.4032.35.007) | Göttingen, Germany | |
| ppp | 0.5, 1.5 | Baro Transmitter | Thies Clima | EC, ECEB, EC-LC |
| | | (Model 3.1157.10.000) | Göttingen, Germany | |
| Ground heat flux, G ($\mathrm{W\,m^{-2}}$) | -0.05 | Hukseflux HFP01 | Hukseflux | ECEB |
| | | | Delft, The Netherlands | |
| Soil temperature, $T_{Soil}$ (°C) | -0.02, -0.05, | DS18B20 | | ECEB, EC-LC |
| | -0.10, -0.25, -0.5 | | | |
| Water vapour mole fraction, $C_{H_2O_v}$ ($\mathrm{mmol\,mol^{-1}}$) | 3.5, 10 | LI-7200 | LI-COR Inc. | EC |
| | | | Lincoln, Nebraska (USA) | |
| Carbon dioxide mole fraction, $C_{CO_2}$ ($\mathrm{\mu mol\,mol^{-1}}$) | 3.5, 10 | LI-7200 | LI-COR Inc. | EC |
| | | | Lincoln, Nebraska (USA) | |

### 2.2.2 Eddy Covariance Energy-Balance (ECEB)

The energy balance at the surface is

$$R_N - G = H + \lambda E + S + Res \tag{1}$$

with net radiation, $R_N$ ($\mathrm{W\,m^{-2}}$), ground heat flux, G ($\mathrm{W\,m^{-2}}$), sensible heat flux, H ($\mathrm{W\,m^{-2}}$), latent heat flux, $\lambda E$ ($\mathrm{W\,m^{-2}}$),

5  soil energy storage term, S ($\mathrm{W\,m^{-2}}$), and energy balance residual, $Res$ ($\mathrm{W\,m^{-2}}$). By convention a turbulent flux towards the atmosphere is defined as positive and a turbulent flux towards the surface is defined as negative. A positive net radiation corresponds to a surplus of radiative energy at the surface and a positive ground heat flux describes a heat transport into the soil.

The energy balance residual, $Res$, per half-hour interval was calculated from Eq. (1) as follows:

10  $$Res = R_N - \lambda E - G - H - S \tag{2}$$

with $\lambda E$ from EC and EC-LC and H from EC.





The $\lambda E$ from ECEB, $\lambda E_{ECEB}$, was calculated as the residual of the net radiation, the ground- and sensible heat flux and the storage term according to Eq. (1)

$$\lambda E_{ECEB} = R_N - G - H - S - Res \tag{3}$$

whilst assuming a fully closed surface energy balance.

Half-hourly evapotranspiration rates in units of $\mathrm{mm\,30\,min^{-1}}$ were calculated from $\lambda E$ as

$$ET = \frac{\lambda E_{ECEB}}{\lambda} \cdot 1800\,s \tag{4}$$

with $\lambda$ the latent heat of vaporization (Dake, 1972)

$$\lambda = (2.501 - 0.00237\,T) \cdot 10^6, \tag{5}$$

with $\lambda$ being dependent on air temperature, T ($^\circ$C).

The soil heat storage term has a major contribution to the unclosed energy balance (Foken, 2008a) and the magnitude of the soil heat storage is comparably larger than the other storage terms, i.e. the photosynthesis flux, the crop enthalpy change, the air enthalpy change, the canopy dew water enthalpy change and the atmospheric moisture change (Jacobs et al., 2008). We used the ground heat flux, G, from the ground heat flux measurements, $G_{HFP}$ ($\mathrm{Wm^{-2}}$), at the sites and calculated the soil heat storage between the soil heat flux plate and the soil layer above following Liebethal and Foken (2007) as

$$G = G_{HFP} + \int\limits_{z=-0.05\,m}^{0\,m} c_v \frac{\partial T}{\partial t} dz \tag{6}$$

The soil heat storage (second term on the right hand side of Eq. (6)) consists of the vertical integral of the change of temperature over time at depth $z = 0.02$ m. $c_v$ is the volumetric heat capacity of the soil, calculated from the soil components, i.e. organic, mineral and water and their respective heat capacities. Soil texture and bulk densities are summarized in Table A3 and were provided by Göbel et al. (2018) and Schmidt et al. (unpublished data). Gaps in soil storage data were filled according

to a multiple linear regression with soil storage as the independent variable, and net radiation and ground heat flux as the dependent variables. The multiple linear regression fitting parameter were derived from records when the soil storage, the net radiation and the ground heat flux were available at the same time.

### 2.2.3   Low-cost eddy covariance (EC-LC)

The EC-LC set-ups comprised of the same ultrasonic anemometer uSONIC3-omni as used for the EC and ECEB set-ups plus a

compact low-cost relative humidity, air temperature and pressure sensor (BME280, BOSCH, Germany, Table 2). Water vapour mole fraction was calculated using measurements of relative humidity, air temperature and air pressure from the low-cost thermohygrometer. The turbulent water vapour fluxes were calculated as the covariance between the vertical wind velocity and the water vapour mole fraction from EC-LC, as per the principle of the eddy covariance method (Baldocchi, 2014). A detailed description and application of the EC-LC set-up for evapotranspiration measurements over AF and MC is given in Markwitz




and Siebicke (2019). Evapotranspiration from EC-LC was neither gap-filled for the methodological comparison nor for the analysis of the energy balance closure due to the risk for new errors and artefacts from the respective gap-filling method.

## 2.3 Gap-filling and energy balance closure adjustment

### 2.3.1 ECEB

For the calculation of annual sums of ET from ECEB data, gaps were filled with the online eddy covariance gap-filling and flux-partitioning tool REddyProc developed at the Max Planck Institute for Biogeochemistry in Jena, Germany (https://www.bgc-jena.mpg.de/bgi/index.php/Services/REddyProcWeb, last access: 19 January 2020). The methods used therein are based on Falge et al. (2001) and Reichstein et al. (2005). We corrected ET from ECEB for the average energy balance non-closure. We estimated the energy balance non-closure during measurement campaigns. Considering the energy balance residual reduces
ET from ECEB.. We used machine learning to estimate the energy balance residuals (Eq. (2)) during times when no campaigns took place. We used the machine learning technique Extreme Gradient Boosting (Chen and Guestrin, 2016; Chen et al., 2019) and predicted the residual energy for both years, 2016 and 2017, at all sites with the R-package xgboost (Chen et al., 2019).

The calculated residual was treated as the dependent variable, whereas the net radiation, the ground heat flux and the sensible heat flux were treated as the independent variables. The model was tested with the data gathered during the campaigns and
divided into a training period and a testing period. At a ratio of 2/3 of training to testing data, we achieved a Pearson correlation coefficient between the testing and predicted data of 0.66. The trained model was then applied to both years, with the net radiation, the ground heat flux and sensible heat flux as input parameters. As a last step the predicted residual was subtracted from half-hourly ET. We assumed that the residual distributes equally to the latent and sensible heat flux, thus subtracted only half of the residual from ET. Commonly, the residual energy is partitioned according to the Bowen ratio (Twine et al., 2000),
which requires direct and continuous measurements of H and LE by EC. We decided for an equal separation of the residual energy because direct LE measurements by EC were not available at our sites. This assumption may cause an overestimation of LE during dry ambient conditions, when the Bowen ratio is high. In contrast, LE is expected to be underestimated during moist ambient conditions, when the Bowen ratio is small. As no campaign on the energy balance closure was conducted at the monocultural agriculture plot of Reiffenhausen, we used the data gathered during the campaign at the AF plot of Reiffenhausen
to train the model and to predict the residual at the MC plot.

### 2.3.2 EC-LC

Half-hourly ET rates from EC-LC were gap-filled with half-hour ET rates from ECEB. The data were corrected for the surface energy balance closure as follows

1. The residual energy was estimated from all available data in 2016 and 2017, following Eq. (2).

2. We used the calculated residual as the dependent variable and the net radiation, the ground heat flux and the sensible heat flux as independent variables to train the same machine learning tool as used for ECEB.



3. The residual was predicted by the trained model; data gaps in the residuals, originating mainly from missing LE caused by data quality checks, were filled with the predicted values.

4. Subsequently, we distributed the residual to the half-hourly ET from EC-LC ($\lambda E^{cor}_{EC-LC}$) and to ET from ECEB used for gap-filling ($\lambda E^{gf}_{ECEB}$) as follows.

$$\alpha = 0.5 \tag{7}$$

$$\lambda E^{cor}_{EC-LC} = \lambda E_{EC-LC} + Res \cdot \alpha \tag{8}$$

$$\lambda E^{gf}_{ECEB} = \lambda E^{gf}_{ECEB} - Res \cdot \alpha \tag{9}$$

## 2.4 Energy balance closure estimation

The energy balance closure (EBC) was quantified in two ways:

1. As the linear regression between the available energy ($R_N$ - G - S), and the sum of the turbulent flux components ($\lambda E$ + H). We used the lmodel2 function from the lmodel2 R-package (Legendre and Oksanen, 2018) to compute the linear regression. The major axis linear regression was applied, which assumes equally distributed errors in both time series. We interpret the slope between the available energy and the sum of the turbulent fluxes as the closure of the surface energy balance. A slope of one and an intercept of zero corresponds to perfect energy balance closure. In the present study both the slope and the intercept were considered as variable.

2. As the energy-balance-ratio (EBR) or also called "instantaneous energy balance closure" (Stoy et al., 2013), thus the closure per half an hour:

$$EBR = \frac{\lambda E + H}{R_N - G - S}, \tag{10}$$

with $\lambda E$ from either EC or EC-LC.

## 2.5 Flux footprint analysis

The spatial coverage and the position of the source area of turbulent sensible- and latent heat fluxes, as well as momentum at a specific point in time is defined by the flux footprint (Schmid, 2002; Kljun et al., 2015). In the present study a flux footprint climatology was calculated with the flux footprint prediction online data processing tool developed by Kljun et al. (2015) (http://footprint.kljun.net/, last access: 19 January 2020). The analyses were performed separately for the respective campaign periods (Table A1 for dates) and for both years at each site. We selected only daytime data, according to a global radiation $R_G > 20\ \mathrm{Wm^{-2}}$.

## 2.6 Canopy resistance

Effects of structural differences between AF and MC on ET were studied in terms of the relationship between the aerodynamic and canopy resistances $(\mathrm{s\,m^{-1}})$ and half-hourly ET. The canopy conductance was calculated using the rearranged Penman-





Monteith equation (11) for evapotranspiration, which depends on the canopy conductance, $g_c$ $(\mathrm{m\,s^{-1}})$, and the aerodynamic conductance for heat, $g_{ah}$ $(\mathrm{m\,s^{-1}})$. The canopy conductance follows the big leaf assumption, assuming that the whole canopy response to environmental changes is equal to the response of a single leaf. The Penman-Monteith equation for evapotranspiration of a canopy (Monteith, 1965) is

$$\lambda E = \frac{s(R_N - G) + c_p\, VPD\, g_{ah}}{s + \gamma(1 + g_{ah}/g_c)} \tag{11}$$

with the vapour pressure deficit, $VPD = e_*(T_a) - e_a$, the heat capacity at constant pressure, $c_p = 1005$ $\mathrm{J\,(kg\,K)^{-1}}$, the saturation vapour pressure (hPa),

$$e_*(T_a) = 0.6112 \exp((17.67 T_a)/((T_a + 273.15) - 29.66)), \tag{12}$$

and the psychrometer constant, $\gamma = (c_p\, ppp)/(\lambda\, 0.622)$. The slope of the saturation vapour pressure curve, $s$, is

$$s = \frac{\varepsilon \lambda q_{sat}}{R_v T_a} \tag{13}$$

with $\varepsilon = 0.622$ and the specific humidity at saturation, $q_{sat} = \varepsilon e_*(T_a)/ppp)$ as a function of temperature.

Rearranging Eq. (11) yields the canopy resistance, $r_c$ $(\mathrm{s\,m^{-1}})$,

$$r_c = \frac{1}{g_c} = \frac{s/\gamma + 1}{g_{ah}} \left[ \frac{s/\gamma(R_N - G)}{(s/\gamma + 1)\lambda E} - 1 \right] + \frac{c_p\, VPD}{\gamma \lambda E} \tag{14}$$

The aerodynamic conductance for heat is

$$g_{ah} = \frac{\kappa^2 u}{\left( \ln\left(\frac{z-d}{z_{0m}}\right) - \psi_m(\zeta) \right)\left( \ln\left(\frac{z-d}{z_{0h}}\right) - \psi_h(\zeta) \right)} \tag{15}$$

with the von Karman constant, $\kappa = 0.4$, the horizontal wind velocity, $u$ $(\mathrm{m\,s^{-1}})$, the measurement height, $z$ (m), the displacement height, $d$ (m), estimated as 70 % of the canopy height, the roughness length for momentum transport, $z_{0m}$, estimated as 10 % of the canopy height and the roughness length for heat transport, $z_{0h}$, estimated as 10 % of $z_{0m}$. $\psi_m(\zeta)$ is the universal function for momentum and $\psi_h(\zeta)$ is the universal function for heat. $\psi_m(\zeta)$ and $\psi_h(\zeta)$ depend on atmospheric stability with the stability parameter $\zeta = (z - d)/L$, including the Monin-Obukhov length, $L$. $\psi_m$ and $\psi_h$ were calculated as

$$\psi_m(\zeta) = \begin{cases} 2\ln[(1+x)/2] + \ln[(1+x^2)/2] & \text{for } \zeta < 0 \\ -2 atan(x) + \pi/2 & \\ -5\zeta & \text{for } \zeta \geq 0 \end{cases} \tag{16}$$

$$\psi_h(\zeta) = \begin{cases} 2\ln[(1+x^2)/2] & \text{for } \zeta < 0 \\ -5\zeta & \text{for } \zeta \geq 0 \end{cases} \tag{17}$$





with x $= (1 - 16\zeta)^{1/4}$ (Bonan, 2016; Businger et al., 1971; Stull, 1989).

We studied the relationship between ET and canopy resistance as well as aerodynamic resistance for idealized ambient conditions, with global radiation, $R_G \geq 400 \text{ W m}^{-2}$, horizontal wind speed, $u \geq 1 \text{m s}^{-1}$ and vapour pressure deficit, $VPD = 1 \pm 0.3 \text{kPa}$ (Schmidt-Walter et al., 2014).

In order to investigate the relationship between the dryness index (potential evapotranspiration divided by precipitation) and the evapotranspiration index (actual evapotranspiration divided by precipitation) we used the simplified Priestley Taylor equation after (Priestley and Taylor, 1972) to calculate a potential evapotranspiration. The Priestley Taylor equation depends only on available energy and air temperature

$$ET_{PT} = \alpha_{PT} \frac{s}{s + \gamma}(R_N - G) \tag{18}$$

with the Priestley-Taylor coefficient, $\alpha_{PT} = 1.25$.

## 3   Results and discussion

### 3.1   Meteorological conditions and plant physiological stages during the campaigns

During the measurement campaign at the MC plot of Dornburg (16 June to 14 July 2016, Fig. 2 a), we observed a mean air temperature of 18.6 °C and a mean vapour pressure deficit (VPD) of 7.35 hPa. High mean air temperature and low rainfall of

2.1 mm over the campaign period led to rapid crop ripening.

At the cropland AF plot of Dornburg (14 July to 12 August 2016, Fig. 2 b) mean air temperature was 19 °C. The measuring period was characterized by frequent precipitation events with total rainfall of 57.1 mm. During this period poplar trees were at the seasonal maximum of their productivity and crops were mature.

During the measurement campaign at the grassland AF plot of Reiffenhausen (12 August to 14 September 2016, Fig. 2 c),

we observed a mean air temperature of 19.31 °C, a mean VPD of 8.02 hPa, and few rain events, accounting for 26.3 mm precipitation over this period. Both trees and grasses were at the maximum of their productivity.

At the AF and MC plot of Wendhausen (03 May to 02 June 2017, Fig. 2 d) the mean VPD was 5.4 hPa and 5.2 hPa, respectively. High precipitation during the campaign (48.6 mm at the AF and 90.7 mm at the MC plot) meant water availability was not limited. The large difference in precipitation measurements was caused by the placement of the rain gauge inside the

AF canopy. Mean daily air temperature was between 10 and 15 °C at the beginning of the campaign, whereas more moderate temperature was observed during the end of the campaign, between 15 and 20 °C. Mean air temperature was 16.6 °C at the AF and 15.5 °C at the MC plot over the entire campaign period.

In contrast, the measurement campaign in Forst (08 June to 08 July 2017, Fig. 2 e) was very warm with mean air temperature of 21.4 °C at the AF plot and 21.2 °C at the MC plot. Mean VPD was 12.02 and 11.88 hPa and precipitation was 18.9 and

14.8 mm at the AF and the MC plots, respectively. The dry and warm conditions initiated the ripening process earlier, resulting in mature crops at the beginning of the campaign.





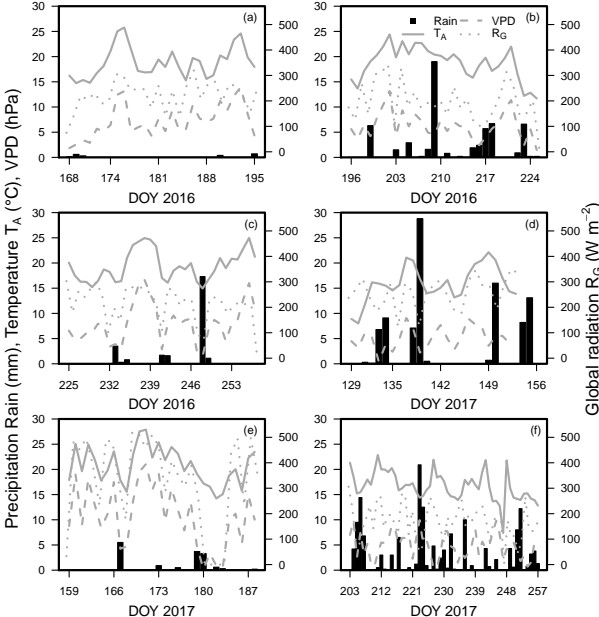

**Figure 2.** Time series of daily mean air temperature, T$_A$, vapour pressure deficit, VPD, daily summed precipitation (left y-axis) and daily mean global radiation, R$_G$, (right y-axis) for Dornburg MC, (a), Dornburg AF, (b), Reiffenhausen AF (c), Wendhausen (d), Forst, (e), and Mariensee, (f). The data for MC and AF of the respective sites of Wendhausen, Forst and Mariensee were averaged. The field campaigns at the AF and MC plots were conducted during the same time and we assumed similar weather conditions due to the small distance between the AF and MC plot.

The last and longest measurement campaign at the grassland AF and MC plot of Mariensee (21 July to 19 September 2017, Fig. 2 f) followed the seasonal trend from high daily mean air temperature in July of 20 °C to lower daily mean air temperature in September at 15 °C. Mean VPD was 6.2 hPa and 4.7 hPa at the AF and MC plots, respectively. Precipitation during this period was 40.6 mm and 163.5 mm at the AF and MC plots, respectively.

## 3.2 Flux footprint climatology

5 The flux footprint analyses showed that the measured turbulent fluxes were representative for the larger AF plots and their respective MC plots during the time of the experiments (e.g. Dornburg, Forst and Wendhausen, Figs. 3 a, c and e). At the AF and MC plots of Dornburg 80 % of the flux magnitude originated from the respective system. The 90 % flux magnitude contribution line at the AF plot overlapped with the 90 % flux magnitude contribution line at the MC plot towards the west. 10 The overlapping footprint was also found for the annual footprint analyses (Fig. A3 a).

At the AF and the MC plot of Wendhausen we observed an 80 % flux magnitude contribution from both land-uses to the total turbulent flux (Fig. 3 e). A 10 % flux contribution originated from the forest around 200 m east of the flux tower. Easterly





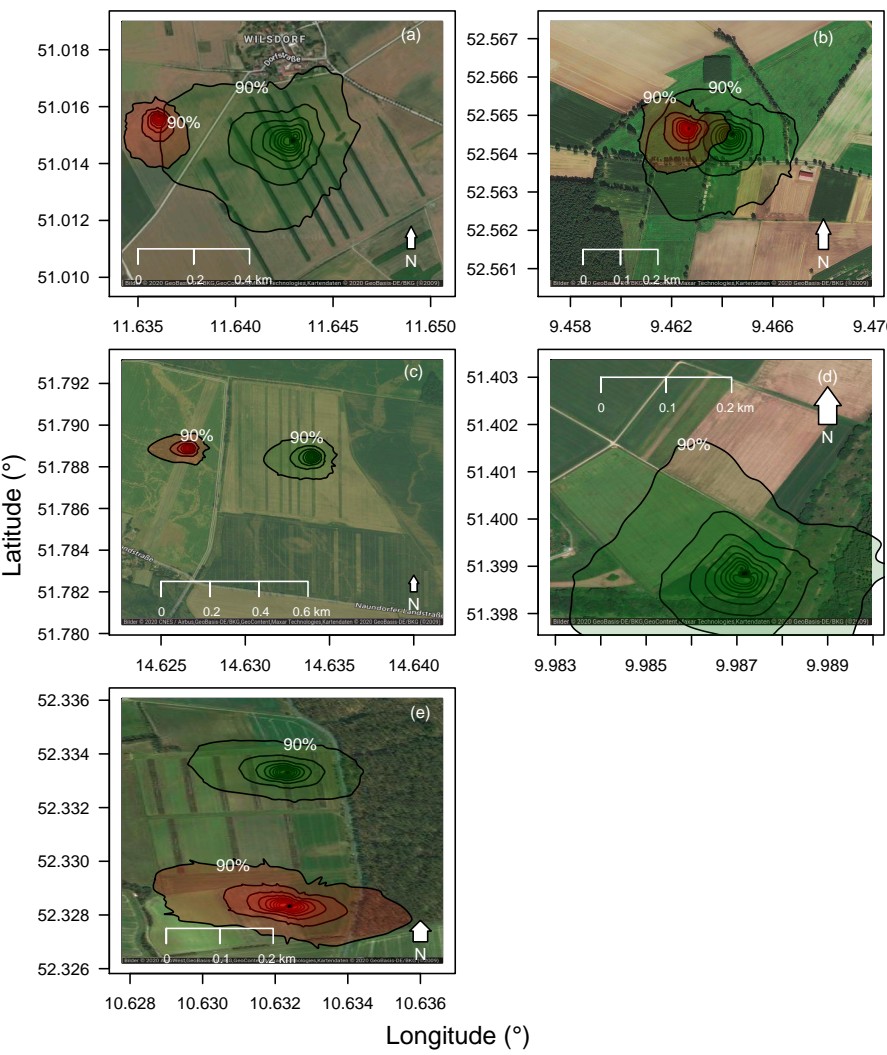

**Figure 3.** Flux footprint climatologies for Dornburg, (a), Mariensee, (b), Forst, (c), Reiffenhausen, (d), and Wendhausen, (e), for the respective campaign period. Green shaded footprints correspond to the AF plot and red shaded footprints correspond to the MC plot. For the analysis only daytime data were used ($R_G > 20\ \mathrm{W m^{-2}}$). Isolines correspond to a 10 to 90 % flux magnitude contribution in 10 % steps, with the 90 % isoline labelled in the plot. Aerial photographs originate from Google maps/ Google earth ©Google 2020.





winds are most likely during stable atmospheric stratification in winter or summer. During the time of the experiment the wind mainly originated from westerly directions (not shown).

70 % of the area of the AF and MC grassland plots of Mariensee contributed to the measured fluxes, respectively (Fig. 3 b). The remaining 20 % of the area contributing to the measured flux originated from surrounding crops and the AF and MC

grassland systems. There was an overlap of the two footprints at the AF and the MC grassland plot, which was expected, as both flux towers are separated by a distance of about 200 m.

The fluxes measured at the smallest AF plot in Reiffenhausen were influenced by fluxes originating from the nearby forests and crop fields about 400 m distance to the flux tower in northerly direction and about 200 m distance in southerly direction (Fig. 3 d). Only 60 % of the fluxes originated from the willow-grassland AF system and the short rotation willow plantation in

the west. The terrain at the AF plot of Reiffenhausen is sloped towards the north-west. The main wind direction at the site was north-northwest in the direction of the sloped terrain.

### 3.3  Diel evapotranspiration

The diel variation of ET for all three set-ups at all sites is depicted in time series plots for an exemplary time period in Figure 4.

The EC-LC set-up showed the best performance relative to direct EC measurements with coefficients of determination between minimum 71% and maximum 94%. The EC-LC set-up captured the temporal variability of ET and the flux response to changing ambient conditions as good as direct EC measurements. The slopes from a linear regression analysis of LE from EC-LC versus EC, showed an agreement between 86% and 99% across four agroforestry plots and between 108% and 142% across four monoculture agriculture plots (Table 3 and Fig. A2).

At the MC plots of Forst and Wendhausen (Fig. 4 d and i) we observed comparably high fluxes obtained by EC-LC relative to direct EC measurements, while attaining high coefficients of determination. We suspect that the laser source of the LI-7200 gas analyser did not work as expected as indicated by spectral analysis (data not shown). Only low-frequency fluctuations were sampled, whereas the high-frequency fluctuations were attenuated. The measured water vapour mole fraction from the gas analyser was similar to the derived water vapor mole fraction from the thermohygrometer of slow response at the respective

sites in terms of magnitude and temporal variability (data not shown). The similarity of the water vapor mole fractions indicate similar spectral response characteristics of the gas analyser and the thermohygrometer set-up. Therefore, the correction of high-frequency losses is expected to be higher for the compromised gas analyser at the respective MC plots, than for a fully functional gas analyser.

ET obtained by the alternative ECEB set-up also captured the diel cycle of ET and gave an indication on the ecosystem

response to changing meteorological driver (Fig. 4). ET was overestimated by ECEB relative to EC across all sites. A minimum overestimation of 27 % was observed at the AF plot of Forst and a maximum overestimation of 101 % was observed at the MC plot of Forst at half-hourly time scale (Table 3 and Fig. A1). Differences between ET from ECEB and EC were attributed to the assumption of a fully closed energy balance at the surface (Foken et al., 2006). ET from ECEB was calculated as the residual



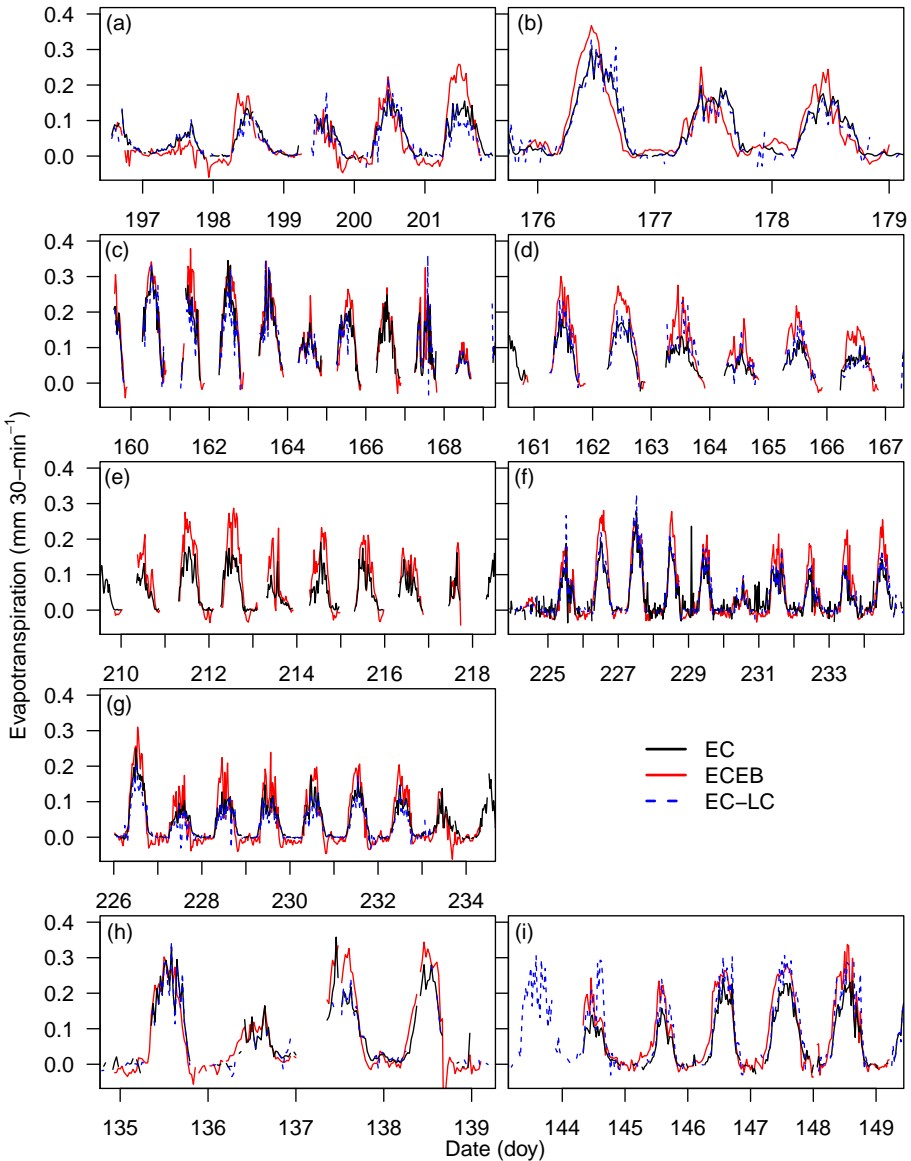

**Figure 4.** Time series of half-hourly evapotranspiration rates of an exemplary time period, for ECEB, EC-LC and EC as a reference, for Dornburg AF, (a), Dornburg MC, (b), Forst AF, (c), Forst MC, (d), Mariensee AF, (e), Mariensee MC, (f), Reiffenhausen AF, (g), Wendhausen AF, (h), and Wendhausen MC, (i). The presented time series were not corrected for the energy balance non-closure.

of net radiation, sensible heat flux, ground heat flux and soil storage. In this analysis we did not account for the commonly observed non-closure of the energy balance and added the surface energy balance residual completely to LE.





**Table 3.** Statistical analysis results for a linear regression of LE from EC-LC versus EC and from ECEB versus EC. Shown are the root mean square error, RMSE, the standard deviation of the differences between both set-ups, SD, the bias, Bias, the number of points used for the analysis, n, the slope for a linear regression of LE from EC-LC versus EC and from ECEB versus EC, as well as the coefficient of determination of the linear regression, $R^2$.

| Sites | Method | RMSE ($Wm^{-2}$) | SD ($Wm^{-2}$) | Bias ($Wm^{-2}$) | n | Slope | $R^2$ |
|---|---|---|---|---|---|---|---|
| Dornburg AF | ECEB/EC | 67.65 | 67.33 | -6.23 | 1202 | 1.93 | 0.45 |
| | EC-LC/EC | 35 | 31.93 | -11.14 | 1037 | 0.94 | 0.71 |
| Dornburg MC | ECEB/EC | 71.53 | 71.51 | 2.31 | 1152 | 1.33 | 0.52 |
| | EC-LC/EC | 34.31 | 34.3 | 1.1 | 1030 | 1.08 | 0.86 |
| Forst AF | ECEB/EC | 58.91 | 57 | 7.64 | 549 | 1.27 | 0.79 |
| | EC-LC/EC | 38.5 | 36.74 | -2.13 | 197 | 0.95 | 0.9 |
| Forst MC | ECEB/EC | 74.5 | 61.70 | 18.42 | 612 | 2.01 | 0.7 |
| | EC-LC/EC | 37.9 | 34.5 | 5.3 | 461 | 1.42 | 0.8 |
| Mariensee AF | ECEB/EC | 79.79 | 65.54 | 23.82 | 1503 | 2.0 | 0.78 |
| | EC-LC/EC | – | – | – | – | – | – |
| Mariensee MC | ECEB/EC | 61.1 | 59.81 | 8.81 | 1852 | 1.42 | 0.75 |
| | EC-LC/EC | 44.6 | 43.9 | 4.62 | 1520 | 1.16 | 0.8 |
| Reiffenhausen AF | ECEB/EC | 55.4 | 55.23 | 4.1 | 1395 | 1.65 | 0.74 |
| | EC-LC/EC | 27.84 | 23.61 | -2.72 | 279 | 0.86 | 0.9 |
| Wendhausen AF | ECEB/EC | 68.30 | 67.88 | 5.34 | 954 | 1.3 | 0.8 |
| | EC-LC/EC | 33.5 | 32.7 | -3.1 | 586 | 0.99 | 0.94 |
| Wendhausen MC | ECEB/EC | 73.42 | 61.14 | 24.4 | 792 | 1.41 | 0.85 |
| | EC-LC/EC | 57.9 | 47 | 15.53 | 604 | 1.3 | 0.89 |



### 3.4 Energy Balance Closure (EBC)

#### 3.4.1 EBC from EC and EC-LC

The EBC ($\pm5\%$ confidence interval) for LE from EC was 82% $\pm2\%$ and 88% $\pm2.5\%$ for Dornburg, 87% $\pm2\%$ and 78% $\pm2\%$ for Forst, 65% $\pm1\%$ and 75% $\pm1.5$ % for Mariensee and 83% $\pm2\%$ and 76% $\pm2\%$ for Wendhausen, for the AF and MC plots,

respectively, and 80% $\pm1\%$ for the AF plot of Reiffenhausen (Fig. 5 and Table 4). The coefficient of determination, $R^2$, was minimum 0.77 and maximum 0.92 across sites (Table 4).

The EBC for LE from EC at the AF and the MC plots were comparable to agricultural systems as reported by Stoy et al. (2013), who found a mean EBC of 84$\pm$20 % across 173 FLUXNET sites, a mean EBC of 91 % to 94 % for evergreen broadleaf forests and savannas and a mean EBC of 70% to 78% for crops, deciduous broadleaf forests, mixed forests and

wetlands. Imukova et al. (2016) found an EBC of 71% and 64% for two consecutive growing seasons over a winter wheat stand in Germany. Studying a belt and alley system in Australia Ward et al. (2012) found an EBC between 67% and 80% over the time period of half a year. Fischer et al. (2018) reported on water requirements of three short rotation poplar stands and found a mean long-term energy balance closure of 82 % at a site in Italy, an EBC of 91 or 95% at a site in the Czech Republic and an EBC of 69% at a site in Belgium. At our sites we found a mean EBC of 79.4% $\pm8.4\%$ across the five AF plots and a

mean EBC of 79.25% $\pm6\%$ across the four MC plots for LE from EC.

The EBC for LE from EC-LC was slightly lower at the AF plots with a mean EBC of 78% $\pm4.24\%$, for five sites, compared to the MC plots with a mean EBC of 81.4% $\pm12.44\%$ for five sites. At the AF plots we observed a lower mean EBC from EC-LC compared to the mean EBC from EC.

The differentiation into lower EBC at the AF and higher EBC at MC plots observed for the two different set-ups is in

agreement with the linear regression results presented in Section 3.3. At the AF plots LE from EC-LC was lower than LE from EC. In the calculation of the energy balance closure only LE was changed and the other energy balance components were held constant. Therefore, increased LE led to a decreased residual energy and subsequently to a better fit of the energy balance closure.

#### 3.4.2 Diel cycles of the energy balance ratio and the energy balance residual

The diel cycle of the energy balance ratio (EBR) from EC at our sites can be classified into two different patterns. The first group of sites (Dornburg, Fig. 6 a, and Wendhausen, Fig. 6 e) show a square diel cycle of the EBR. The EBR is minimum 0 at 6 am and maximum 1.8 at 6 pm. The diel cycle of the EBR at the second group of sites (Forst, Fig. 6 b, Mariensee, Fig. 6 c, and Reiffenhausen, Fig. 6 d) is lowest at 6 am and 6 pm with an EBR of 0.5, whereas between 8 am and 4 pm the EBR is fairly constant at a level of 0.80 at Forst, 0.65 at Mariensee and 0.75 at Reiffenhausen. The mean EBRs are in the same

range as the EBC estimated for all sites and the whole campaign. Sites with the first group (Dornburg and Wendhausen) might be affected by horizontal advection of heat and moisture. Oncley et al. (2007) reported that the advection of moisture had the highest contribution to the unclosed energy balance compared to the other components and the maximum peak of the horizontal moisture advection term was in the afternoon, as energy was accumulated during the day and released in the afternoon.



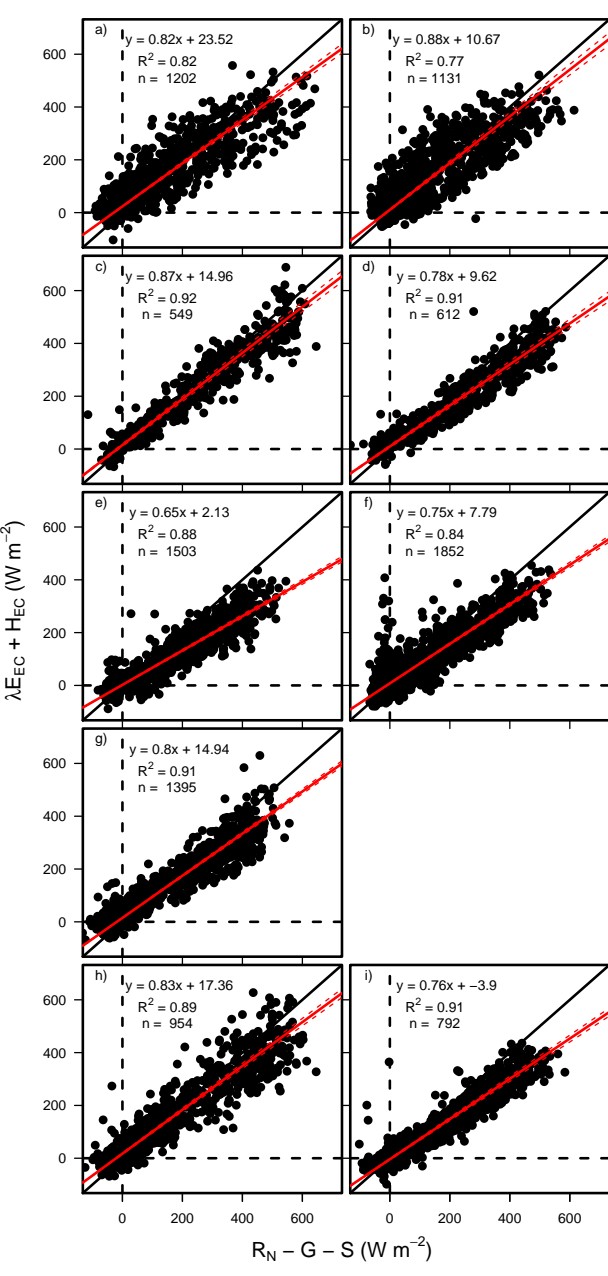

**Figure 5.** Scatterplot of the sum of the turbulent fluxes ($\lambda E_{EC} + H_{EC}$) versus the sum of the available energy ($R_N - G - S$) for Dornburg AF, (a), Dornburg MC, (b), Forst AF, (c), Forst MC, (d), Mariensee AF, (e), Mariensee MC, (f), Reiffenhausen AF, (g), Wendhausen AF, (h), and Wendhausen MC, (i). The plot contains the linear regression equation, the coefficient of determination, $R^2$, and the number of data points used for the analysis, n.





**Table 4.** Statistical analysis results of the linear regression between the sum of the turbulent fluxes and the available energy. Namely, the sites, the set-up used, the slope ($\pm 5\%$ confidence interval), intercept, the coefficient of determination of the linear regression, $R^2$, and the number of points used for the analysis, n. The energy balance closure determined by EC-LC at Mariensee AF is based on data collected from 23 March 2016 to 20 November 2016 and at Reiffenhausen MC the analyses are based on data collected from 07 April 2016 to 31 December 2016, because no data were available during the campaigns.

| Sites | Set-up | Slope | Intercept ($\mathrm{Wm}^{-2}$) | $R^2$ | n |
|---|---|---|---|---|---|
| Dornburg AF | EC | $0.82\pm 0.02$ | $23.52\pm1.95$ | 0.82 | 1202 |
| | EC-LC | $0.75\pm 0.03$ | $17.01\pm2.6$ | 0.72 | 1088 |
| Dornburg MC | EC | $0.88\pm 0.025$ | $10.67\pm3.1$ | 0.77 | 1131 |
| | EC-LC | $0.90\pm 0.035$ | $10.58\pm4.0$ | 0.71 | 1046 |
| Forst AF | EC | $0.87\pm 0.02$ | $14.96\pm5.1$ | 0.92 | 549 |
| | EC-LC | $0.81\pm 0.045$ | $17.17\pm11.1$ | 0.85 | 205 |
| Forst MC | EC | $0.78\pm 0.02$ | $9.66\pm4.4$ | 0.91 | 612 |
| | EC-LC | $0.85\pm 0.03$ | $10.28\pm7.9$ | 0.85 | 486 |
| Mariensee AF | EC | $0.65\pm 0.01$ | $2.13\pm1.63$ | 0.88 | 1503 |
| | EC-LC | $0.81\pm 0.009$ | $1.7\pm0.6$ | 0.83 | 6574 |
| Mariensee MC | EC | $0.75\pm 0.015$ | $7.8\pm1.2$ | 0.84 | 1852 |
| | EC-LC | $0.82\pm 0.015$ | $7.7\pm1.4$ | 0.88 | 1632 |
| Reiffenhausen AF | EC | $0.80\pm 0.01$ | $14.94\pm1.2$ | 0.91 | 1395 |
| | EC-LC | $0.72\pm 0.03$ | $10.55\pm3.1$ | 0.91 | 306 |
| Reiffenhausen MC | EC | – | – | – | – |
| | EC-LC | $0.60\pm 0.006$ | $6.02\pm0.37$ | 0.83 | 9621 |
| Wendhausen AF | EC | $0.83\pm 0.02$ | $17.36\pm2.8$ | 0.89 | 954 |
| | EC-LC | $0.81\pm 0.03$ | $14.3\pm4.4$ | 0.84 | 641 |
| Wendhausen MC | EC | $0.76\pm 0.02$ | $-3.9\pm2.6$ | 0.91 | 792 |
| | EC-LC | $0.90\pm 0.025$ | $3.11\pm4.4$ | 0.86 | 710 |





In addition to advective transport, the unclosed surface energy balance could be related to energy storage terms such as biomass, the air or photosynthesis, that have previously not been considered. The pattern seen at Dornburg and Wendhausen may be attributed to a release of energy during the afternoon, which correspond to a surplus of energy and a better closure of the energy balance. In the morning hours the storage terms have an opposite sign, which correspond to a lack of energy and a
subsequent poorer energy balance closure.

Interestingly, the diel pattern of the EBR from EC at both land-uses at all sites are equal. Additionally, the differences between the median diel cycle EBRs (between 6 am and 6 pm) at the AF and the MC plot were small, with differences of 0.032 at Dornburg, 0.13 at Forst, -0.09 at Mariensee and 0.13 at Wendhausen. At Reiffenhausen the EBR was only available for the AF plot. Positive differences indicate higher EBRs at the AF compared to the MC plots and vice versa. The median differences
at the respective sites account for 3.3 % and 3.6 % of the median diel EBR at the AF and the MC plots of Dornburg, for 14.0 % and 16.7 % at Forst, for 14.3 % and 12.7 % at Mariensee and 15.5 % and 17.3 % at Wendhausen, respectively. As both flux towers located at the AF and the MC system at one site are separated by approximately 100 to 500 m and the diel patterns look similar, we suspect that the non-closed surface energy balance at one site is caused by local effects of longer wavelength than the commonly applied averaging period of 30 minutes.
At the Dornburg and Wendhausen sites, the residual energy from EC, i.e. the differences between the available energy and turbulent fluxes, showed positive values during the morning and negative values during the night and transition times of sunrise and sunset (Fig. 6 a and e). We interpret the unequally distributed EBR as caused by unaccounted storage terms with a loss of energy in the morning and a release of energy in the afternoon. At the remaining sites (Forst, Fig. 6 b, Mariensee, Fig. 6 c, and Reiffenhausen, Fig. 6 d) the diel cycle of the residual energy was highest during midday, whereas during the morning and the
evening the residual energy was constant at around zero.

The diel cycles of the EBRs and the residuals were similar for both EC-LC and EC set-ups (Fig. A4). This is promising, as it indicates first, a performance of EC-LC comparable to EC, and, second, the capability of the EC-LC set-up to capture site-specific effects. Nevertheless, the observed differences between EBRs and residuals at the AF and MC at one site were mostly attributed to differences in LE. Higher LE from EC-LC than LE from EC led to higher EBRs.

**3.4.3 EBC sensitivity analysis**

In this section, we investigate the response of the surface energy balance closure to changing energy balance components, e.g. the net radiation, latent-, sensible- and ground heat flux. The results should help to conclude on recommendations for the best installation of an eddy covariance flux tower inside agroforestry systems. For an estimate of the energy balance closure sensitivity to variation in the energy balance components, we changed the magnitude of each energy balance component by
±20 % separately for each site. Subsequently, we calculated a median energy balance ratio for each energy balance component across all sites.

We observed a decrease of the median energy balance ratio by 1.7 %, 6.6 % and 12.4 % if the magnitude of ground-, the sensible- and the latent heat fluxes were decreased by 20 %. For an increase of the flux magnitudes by 20 % we observed an increase of the median energy balance ratio of 3.0 %, 8.3 % and 13.0 % (Fig. 7 a-c). For the net radiation we observed

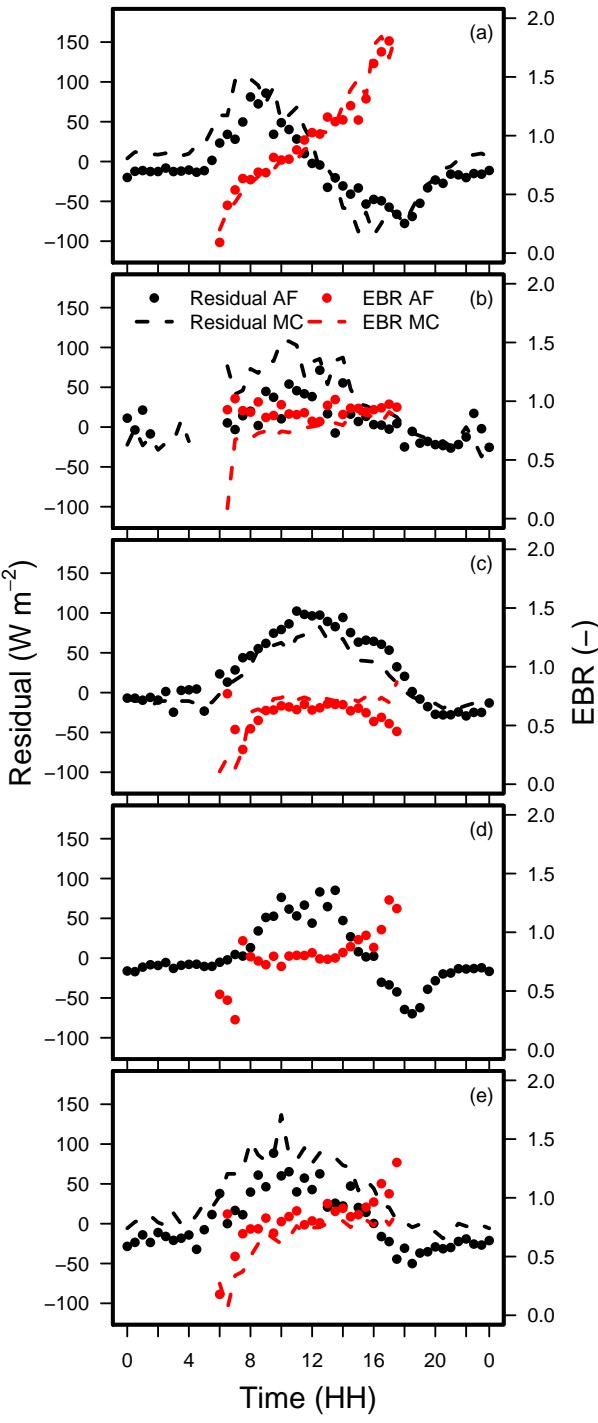

**Figure 6.** Daily median of the energy balance ratio (EBR) and the residual energy at Dornburg, (a), Forst, (b), Mariensee, (c), Reiffenhausen, (d), and Wendhausen, (e), for the AF and the MC plots, respectively. LE and H were obtained by EC.





the opposite, a decrease of the net radiation by 20 % led to an increase of the energy balance closure by 29.2 %, whereas an increase by 20 % led to a decrease of the energy balance closure by 17.73 % (Fig. 7 d).

A change in the median EBR for increased or decreased energy balance components was always attributed to a change of the residual energy. In our study the soil storage term led to an increase in energy balance closure of 1 % to 6 % across sites. Despite the small effect of ground heat flux on EBR, the effect of unrepresentative ground heat flux measurements and the neglection of the soil storage term is not trivial (Foken, 2008a).

In homogeneous surfaces with low vegetation, ground heat flux and soil storage measurements are more representative for the ecosystem of interest than inside heterogeneous AF systems. Wilson (2002) found a phase-shift in the diel cycle of the ground heat flux relative to net radiation and turbulent fluxes inside heterogeneous AF systems. Those phase-shifts occur due to changing incident radiation at the surface, depending on the time of the day and the location of the heat flux plate. One solution for preventing those phase-shifts is to increase the number of heat flux plates and vertical soil temperature profiles. We distributed the heat flux plates and vertical soil temperature profiles in a horizontal transect perpendicular to the crop fields. This made it possible to measure the soil heat flux and soil storage at the two transition zones between crops and trees and within the tree strips.

Compared to the other energy balance components, net radiation had the strongest impact on a change of the EBR. Representative net radiation measurements are difficult to achieve over heterogeneous surfaces such as agroforestry systems. Depending on the placement of the whole measuring complex, either only above trees or above crops or at the transition zone between both trees and crops, the representativity and the magnitude of the measured net radiation varies. The only difference in net radiation for different plants originates from the reflected short wave radiation and emitted long wave radiation. To achieve a representative net radiation over AF we placed the tower on the edge between trees and crops, ensuring the net radiation was representative of that experienced by both tree and crop components.

### 3.5 Evapotranspiration over agroforestry

### 3.5.1 Weekly sums of evapotranspiration

The annual cycle of evapotranspiration across all sites and for the years, 2016 and 2017, depict the typical seasonal cycle of highest ET during summer and lowest ET during winter (Fig. 8). We found small differences between weekly sums of ET at the AF and the MC plots during the main growing period of the crops. After ripening of the crops, we found higher weekly sums of ET at the AF plots compared to the MC plots at the cropland sites of Dornburg (Fig. 8 a), Forst (Fig. 8 b) and Wendhausen (Fig. 8 e). After the ripening of the crops evaporation contributed the most to the measured ET at the MC plot, whereas at the AF plot both evaporation from the crop fields between the tree strips and transpiration from the trees contributed to the measured flux. At the grassland sites (Fig. 8 c and d) differences in weekly sums of ET between both land-uses were small with a tendency of higher ET rates at the MC plot compared to the AF plot.



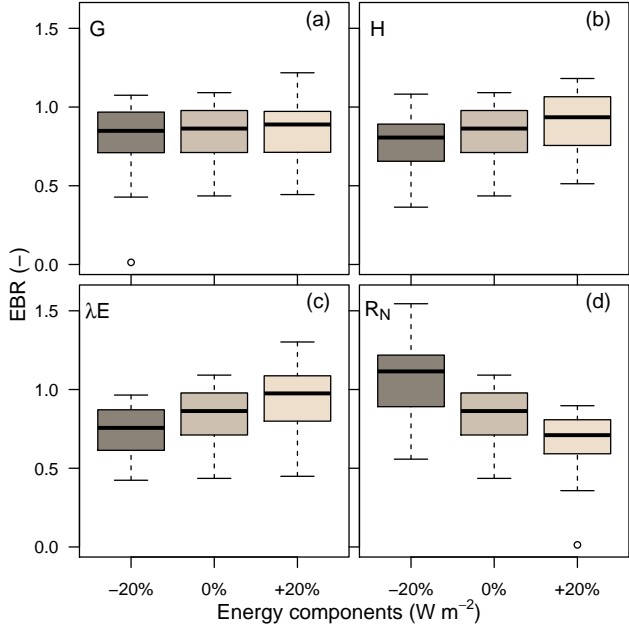

**Figure 7.** Daily median energy balance ratio for a variable ground heat flux, (a), a variable sensible heat flux, (b), a variable latent heat flux, (c), and a variable net radiation, (d), for unbiased values (0 %), for 20% increased and 20 % decreased values (+20% and -20%, respectively). Only daytime data were used with a global radiation $R_G > 20\,\mathrm{W\,m^{-2}}$.

### 3.5.2 Annual sums of evapotranspiration

We compared annual sums of ET for both land uses, AF and MC, both set-ups, ECEB and EC-LC, and both years, 2016 and 2017 (Fig. 9).

In 2016, annual sums of ET from ECEB were higher over AF than over MC in Dornburg, Forst, Reiffenhausen and Wend-
5  hausen (6%, 21%, 10% and 10%) and lower in Mariensee by 16%. This was also found in 2017 for all sites (+31% in Forst, -14% in Mariensee, +20% in Reiffenhausen and +11% in Wendhausen), except for Dornburg, where we observed a slightly lower ET over AF than over MC of 8% (Fig. 9 a and b, and Table 5).

Annual sums of ET from EC-LC in 2016 were higher over AF than over MC in Forst, Mariensee and Reiffenhausen (17%, 14% and 7%, respectively), and slightly smaller over AF than over MC in Dornburg and Wendhausen (1% and 5%). In 2017,
10  the annual sum of ET was higher over AF than over MC in Forst by 6% and vice versa in Wendhausen by 4% (Fig. 9 c and d, and Table 5).

As shown, we only found little differences between annual sums of ET over AF and MC at our sites. Therefore, we wanted to understand how ET over the two land uses (AF and MC) respond to different local climatic conditions. For this purpose we used the relationship between the evapotranspiration index ($\sum\mathrm{ET}/\sum\mathrm{P}$) and the aridity index ($\sum\mathrm{ET}_p/\sum\mathrm{P}$) proposed by

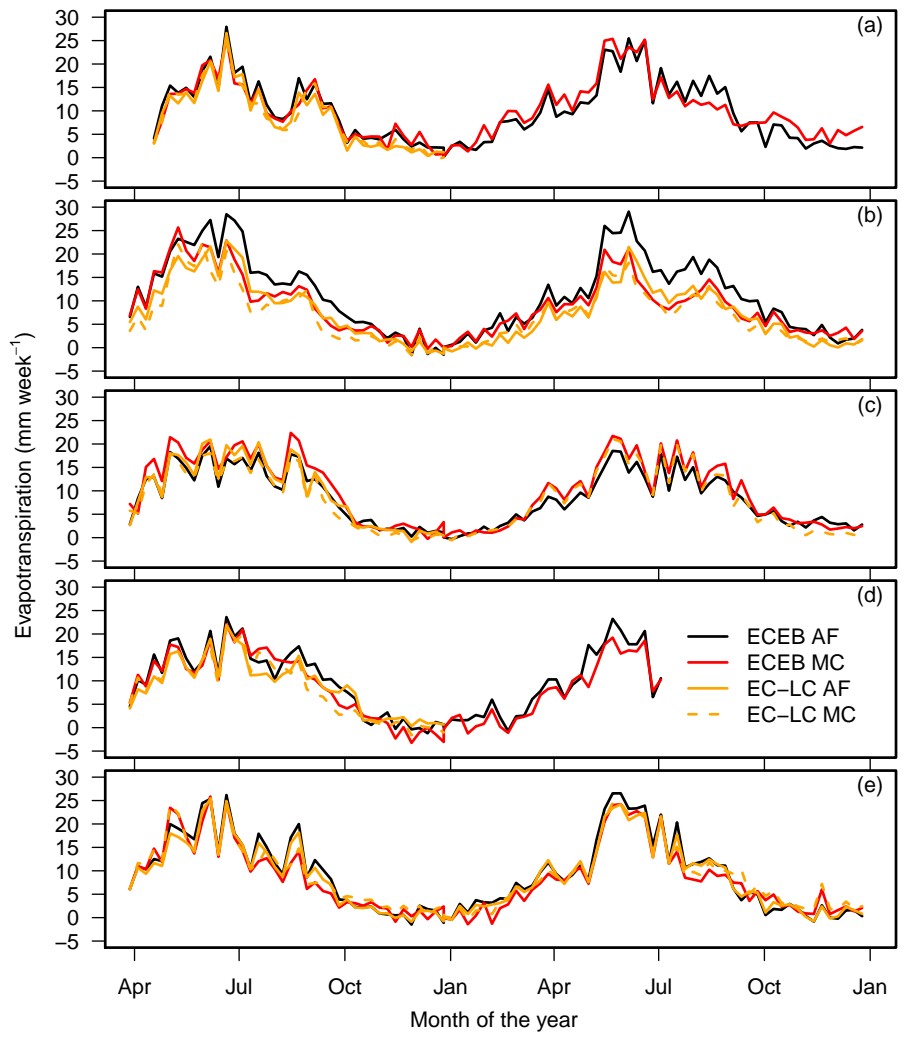

**Figure 8.** Weekly sum of half-hourly ET rates from ECEB (black and red solid lines for AF and MC, respectively) and EC-LC (orange solid and dashed line for AF and MC, respectively) for Dornburg, (a), Forst, (b), Mariensee, (c), Reiffenhausen, (d), and Wendhausen, (e).





Budyko (Budyko, 1974). The local climatic conditions are considered in the calculation of the potential ET (Eq. 18), as per the available energy and air temperature.

Figure 10 shows the ET index as a function of the aridity index for all sites, both set-ups and both years. The figure indicates first that those plots related to an ET index larger than one were water limited, corresponding to an aridity index $ET_p/P>1$.

Secondly, the figure shows a separation of the sites with an energy limitation ($ET_p/P<1$) and water limitation ($ET_p/P>1$) for the years 2016 and 2017, respectively.

In 2016 the grassland sites Mariensee (AF and MC) and Reiffenhausen (AF) had an ET index larger than one. At those sites, the annual sum of ET was generally high relative to the annual sum of precipitation (Fig. A5 c). This finding seems to be typical for grasslands. Williams et al. (2012) reported on average 9% higher transformation of precipitation into evapotranspiration of

grasslands compared to forests across 167 sites as part of the global FLUXNET flux measurement network. They concluded, first, that higher ET of grasslands may have been caused by the less conservative water use compared to trees and, second, that it could indicate that grasses have an extensive, well developed rooting system, similar to trees. Nevertheless, considering the water balance equation with precipitation equalling the sum of evapotranspiration and water runoff, an ET index larger than one indicates water losses via ET and no runoff. An ET index larger than one is only to be expected under ground water access,

irrigation or the impact of a nearby stream. At the grassland site of Mariensee it is likely that the trees and grasses had ground water access, as the ground water table was at about 1.5-2 m depth. The agroforestry system in Reiffenhausen is located on a gentle slope with no ground water access, which we expect should promote run-off, contrary to the high ET index observed. But, the ET measurements are affected by a poplar and willow SRC in the south-southeast and north-northwest directly within the flux footprint (Section 3.2 and Fig. 3). And with respect to the overall area of the agroforestry system, the area covered by

trees amounts to 72% and is much higher, compared to the other sites (Table 1). In both cases, an aridity index larger than one is also possible, despite this indicating a water limitation at the particular sites. Additionally this also indicates a surplus of radiative energy, which promotes photosynthesis and higher transpiration, if water is not limited. In contrast, the Mariensee and Wendhausen sites had evapotranspiration and dryness indices of approximately 0.5 and 0.6 in 2017. Those sites were affected by exceptionally high annual precipitation events, but annual sums of ET comparable to 2016 (Table 5).

The second finding gives evidence for a strong dependency of ET on the local climate. The years 2016 and 2017 correspond to a dry and a wet year, respectively. In Figure 10, arrows indicate the difference between mean evapotranspiraion indices and mean dryness indices grouped by year, method and land-use. The length of the arrows correspond to the overall importance. As shown, the effect of different years with varying hydrological and weather conditions had the strongest impact on mean annual sums of ET, indicated by the movement of points from a water limited (2016) regime to an energy limited (2017) regime.

Differences between mean ET from the two methods had the second largest impact on annual sums, with a trend of higher mean annual sums of ET obtained by ECEB compared to EC-LC. Land-use type had the least impact on ET, with a small trend of higher ET over AF than over MC.

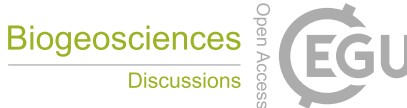

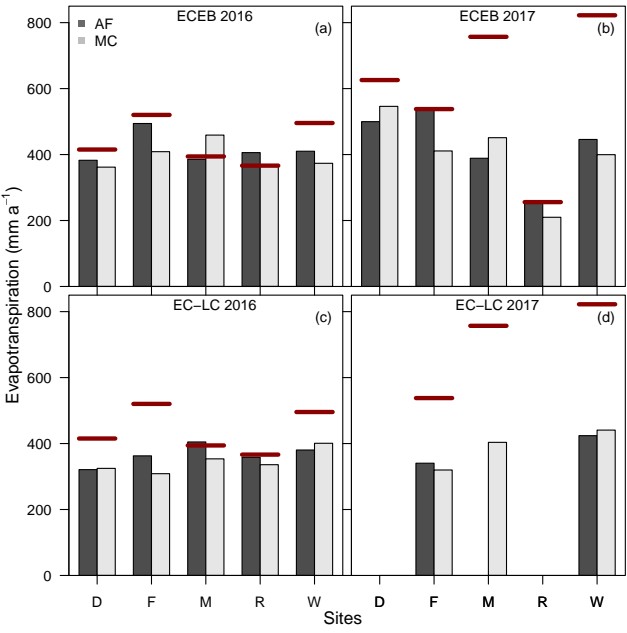

**Figure 9.** Annual sums of evapotranspiration from ECEB in 2016, a, and 2017, b, and from EC-LC in 2016, c, and 2017, d, for Dornburg, "D", Forst, "F", Mariensee, "M", Reiffenhausen, "R", and Wendhausen, "W". The red solid lines correspond to the annual sum of precipitation from the monocultural agriculture plot of the respective site. The annual sums of evapotranspiration at Reiffenhausen AF and MC in 2017 contain only data from 01 January 2017 to 09 July 2017.

### 3.5.3 Effect of agroforestry on ET as explained by aerodynamic and canopy resistance

We wanted to understand if the heterogeneity of the AF systems can explain differences between half-hourly ET rates from AF relative to MC systems. We quantified the effect of heterogeneity on ET as per the relationship between the aerodynamic and canopy resistances and half-hourly ET rates. Tree strips oriented perpendicular to the prevailing wind direction significantly

5    reduced the wind speed (Böhm et al., 2014) and the aerodynamic resistance (Lindroth, 1993). The canopy resistance depends linearly on the aerodynamic resistance and is part of the first term of Eq. (14). If the first term on the right hand side of Eq. (14) is high, the canopy resistance is high and evapotranspiration is controlled by atmospheric processes. Whereas if the aerodynamic resistance is low the second term on the right hand side of Eq. (14) dominates, i.e., ET is mainly controlled by the plants physiology.

10    Mean aerodynamic resistances were lower at the AF plots compared to the MC plots (Fig. 11). We derived aerodynamic resistances with LE from EC-LC. In detail, $r_{ah}$ was lower at the AF compared to the MC plot by 46.4 % at Dornburg, 46.1 % at Forst, 57.7 % at Mariensee, 44.1 % at Reiffenhausen, and 50 % at Wendhausen. The magnitude of $r_{ah}$ at both the AF and the MC plots were comparable to agricultural fields and short rotation plantations (Lindroth, 1993; Schmidt-Walter et al., 2014). Our observations agree with our expectations and the higher roughness incurred by the higher tree alleys led to a decreased





**Table 5.** Annual sums of energy balance closure corrected actual evapotranspiration, ET, potential evapotranspiration, $ET_0$, and precipitation, Rain, $(\mathrm{mm\,a^{-1}})$ for all sites, both set-ups (ECEB and EC-LC) and both years (2016 from April to December, and 2017 from January to December). We included annual sums of uncorrected actual evapotranspiration obtained by ECEB in brackets. The annual sums of ET and precipitation at Reiffenhausen for AF and MC in 2017 contain data from 01 January 2017 to 01 July 2017.

| Method | ECEB | | EC-LC | | | | | |
| Sites | ET 2016 | ET 2017 | ET 2016 | ET 2017 | $ET_0$ 2016 | $ET_0$ 2017 | Rain 2016 | Rain 2017 |
| --- | --- | --- | --- | --- | --- | --- | --- | --- |
| Dornburg AF | 383 (373) | 500 (494) | 321 | – | 523 | 612 | 414 | 626 |
| Dornburg MC | 362 (307) | 546 (511) | 325 | – | 456 | 594 | 414 | 626 |
| Forst AF | 494 (443) | 540 (453) | 363 | 340 | 597 | 590 | 520 | 538 |
| Forst MC | 409 (435) | 411 (397) | 309 | 320 | 589 | 550 | 520 | 538 |
| Mariensee AF | 386 (457) | 389 (433) | 405 | – | 536 | 498 | 394 | 757 |
| Mariensee MC | 459 (466) | 451 (430) | 354 | 404 | 536 | 467 | 394 | 757 |
| Reiffenhausen AF | 406 (390) | 252 (260) | 358 | – | 512 | 330 | 366 | 256 |
| Reiffenhausen MC | 368 (414) | 210 (266) | 336 | – | 534 | 302 | 366 | 256 |
| Wendhausen AF | 410 (417) | 446 (424) | 380 | 424 | 553 | 536 | 496 | 822 |
| Wendhausen MC | 373 (490) | 400 (492) | 401 | 440 | 609 | 547 | 496 | 822 |

aerodynamic resistance at the AF plot compared to the MC plot. Exemplary, we derived an aerodynamic resistance for two different canopy heights of 1 m and 5 m. We assumed a constant wind speed, $u = 2\,\mathrm{m\,s^{-1}}$, universal constants for momentum $\psi_m = 0.9$ and heat $\psi_h = 0.4$, a measurement height z of 10 m and a displacement height d of 7 m. We derived a roughness length for momentum and heat of 0.5 m and 0.05 m for a canopy height of 5 m and 0.1 and 0.01 m for a canopy height of 1 m.

Subsequently, we arrived at an aerodynamic resistance of $10.3\,\mathrm{s\,m^{-1}}$ for a canopy height of 5 m and $41.5\,\mathrm{s\,m^{-1}}$ for a canopy height of 1 m. Thus, a decrease in canopy height of 4 m led to an increase in aerodynamic resistance of 75.2 %.

The relationship between half-hourly evapotranspiration rates and the canopy resistance at our sites followed an exponential function (Fig. 11). The differences between the mean canopy resistances at the AF and the MC plots were much smaller than differences in mean aerodynamic resistances at the AF and the MC plots. We found higher mean canopy resistances at the AF

than at the MC plot of 2.8 % at Dornburg and 7.4 % at Mariensee. Lower mean canopy resistances at the AF than at the MC plot of 5.1 % were found at Forst, 3.7 % at Reiffenhausen and 42 % at Wendhausen. This suggests that the AF and the MC systems behave in a similar way from a plant physiological point of view, regarding the stomatal control of both the trees and the crops. The small differences in canopy resistance at the two land-uses are a good indication for small differences between half-hourly evapotranspiration rates at the AF and the MC plots. Annual sums of ET (from EC-LC for 2016) across sites for

both land-uses were $365\pm28\,\mathrm{mm\,a^{-1}}$ for AF sites and $345\pm39\,\mathrm{mm\,a^{-1}}$ for MC sites and were not significantly different (p = 0.354). Annual sums of ET averaged for the years 2016 and 2017 were also not significant (p=0.7) with a mean annual sum of ET of $370\pm38\,\mathrm{mm\,a^{-1}}$ for AF sites and $361\pm51\,\mathrm{mm\,a^{-1}}$ for MC sites. We conclude that the cultivation of fast growing trees and crops on the same piece of land does not have a significant effect on annual sums of ET, rejecting our initial hypothesis.

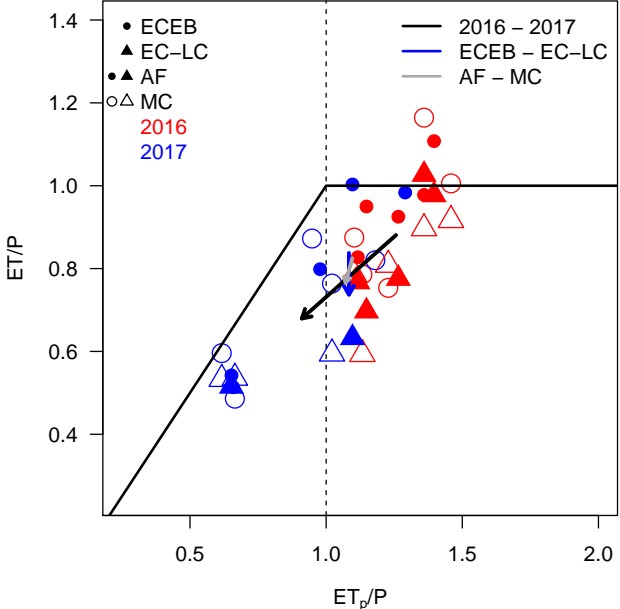

**Figure 10.** Evapotranspiration index ($\sum ET/\sum P$) versus the dryness index ($\sum ET_p/\sum P$) for both land-uses (AF: filled triangles and dots; MC: empty triangles and dots), both set-ups (ECEB: dots; EC-LC: triangles) and both years (2016: red; 2017: blue). The bold black line describe regions of an energy limitation ($ET_p/P<1$) and a water limitation ($ET_p/P>1$). The arrows indicate mean trends of ET for the effect of different years (black arrow), different methods (blue arrow) and different land-uses (grey arrow).

The agroforestry systems investigated in the current study had a sustainable water-use strategy, indicated by small differences in the annual sums of ET between AF and MC sites. Effects of agroforestry on evapotranspiration rates are mostly attributed to a small region next to the tree strips (Kanzler et al., 2018), the quiet zone. There, the reduction of wind velocity and incident radiation is strongest and this causes a reduction of evapotranspiration. The quiet zone extends to roughly 4 to 12 times the

tree height (Nuberg, 1998). The quiet zone changes to the wake zone, where the wind velocity increases and light is no longer limited, hence, evapotranspiration increases towards the centre between tree strips (Kanzler et al., 2018). As a result, both positive and negative effects of trees on evapotranspiration might compensate each other. A similar effect occurs when evapotranspiration is measured over a whole agroforestry system with e.g. the EC method (Baldocchi, 2003). EC measurements integrate over a larger area and small scale differences attributed to the quiet zone can not be detected. Hence, differences can

be small.

The effect of trees on wind velocity and evapotranspiration reduction is strongest for shorter tree strip distances, hence, a high relative area covered by trees (Böhm et al., 2014; Nuberg, 1998). At our sites, the relative area covered by trees is small for the larger sites (e.g. Dornburg, Forst, Mariensee and Wendhausen) (Table 1) and varies between 6 and 12 %. The distances between the tree strips varies between 24 and 96 m. Böhm et al. (2014) argued that a distance larger than 50 m is already

too wide for efficient wind velocity protection and the associated evapotranspiration reduction. For larger distances between





tree strips, the generated turbulent kinetic energy within the AF system might be equal or even larger than at the MC system (McNaughton, 1988), which causes higher ET over AF relative to a MC system. At the smallest site ($\approx 2\,\mathrm{ha}$), Reiffenhausen, the relative area covered by trees amounts to 72 %. At this site our measurements reveal slightly lower annual sums of ET over AF relative to MC (Fig. 9). The AF system is in the south-west and north-west limited by a poplar and willow short

rotation coppice plantation, directly within the footprint of the tower (Fig. 3 d). The observations from Reiffenhausen agree with investigations on water use of short rotation coppices in the Czech Republic (Fischer et al., 2013) and in Belgium (Zenone et al., 2015). Hence, the AF system in Reiffenhausen seems to behave more like a SRC.

### 3.6   Uncertainty and limitations of ET measurements over AF

As outlined in the previous section, differences in annual sums of ET between the different land-uses were small and not

significant. Besides the discussed ecological reasons, we are aware of measurement errors due to the heterogeneous terrain (Foken, 2008b). The most critical assumptions of the eddy covariance method are horizontally homogeneous terrain and steady state ambient conditions (Foken et al., 2006; Foken, 2008b). It is assumed that the heterogeneities generate turbulent motions of longer time scale than the commonly applied averaging period of half an hour. This is also strongly connected to horizontal advection, commonly not properly represented in eddy covariance flux measurements. Foken et al. (2006) noted that the eddy

covariance method is the most accurate method with errors between 5 and 10 %, depending on the turbulent conditions. The errors are higher during nighttime, due to limited turbulent conditions, causing a common flux underestimation (Aubinet et al., 2010). But during night especially ET is small and the effect of high errors are small, compared to daytime conditions when ET is high.

For our low-cost eddy covariance set-up we anticipate higher errors compared to direct EC, due to the limited time response

of the thermohygrometer and subsequently higher spectral correction factors (Markwitz and Siebicke, 2019). We found that the effect of heterogeneity on ET is less important for EC-LC than the effect of different measurement heights (Markwitz and Siebicke, 2019). For a measurement height of 3.5 m, we found a flux underestimation compared to direct EC, and for a measurement height of 10 m, we found a slight flux overestimation (Table 3). At lower height the contribution of small and high-frequency fluctuations to the energy spectrum is higher. Due to the limited time response of the thermohygrometer

between 1.9 and 3.5 seconds (Markwitz and Siebicke, 2019), the high-frequency eddies can not be adequately detected and the signal losses are higher.

In contrast, ET obtained by the ECEB set-up might be affected by greater errors than EC-LC, due to multiple error sources inferred from each of the energy balance components, the assumption of a fully closed energy balance and resulting inaccuracies from the energy balance residual partitioning. For ET obtained by ECEB the heterogeneity of the landscape has a larger

impact than for the EC-LC set-up, such as net radiation and ground heat flux measurements are not representative for the whole landscape (Section 3.4.3 and Figure 7).

Although errors for ET measurements with the respective set-ups can be large on a half-hourly time scale, for annual sums of ET, the errors often compensate each other and are small in relation to the measured signal (Hollinger and Richardson, 2005). As an example, we calculated the random error uncertainty after Hollinger and Richardson (2005) for latent heat fluxes
from Dornburg AF for 2016. The larger the integration time (hourly, daily and monthly), the smaller the random error. The magnitude of the random error was about 2.3 % (median over n = 9) of the flux magnitude for monthly averages, 11.55 % (n = 254) for daily averages and 34.5 % (n = 12191) for hourly averages. Hence, the random error for annual sums would be even smaller.

**4 Conclusions**

The main objective of the current work was to investigate the effect of agroforestry on evapotranspiration in comparison to conventional agriculture. We performed ET measurements at multiple sites, requiring methods of low cost and low maintenance. Therefore we measured evapotranspiration for two consecutive years obtained by a low-cost eddy covariance set-up and an eddy covariance energy balance set-up.

In the first part of this work we investigated the performance of the measurement set-ups. In comparison with direct eddy covariance measurements the low-cost eddy covariance set-up captured the temporal variability in half-hourly ET rates with high coefficients of determination during a comparison measuring campaign. The ECEB set-up also represented the diel cycle of ET, but was characterized by more scatter. We therefore conclude that the EC-LC set-up is a viable alternative compared to conventional eddy covariance set-ups, as the set-up represents ET of the underlying ecosystem more accurately than the ECEB

set-up.

   In the second part of this work we focused on the question if agroforestry systems have higher water losses to the atmosphere via ET compared to monoculture systems. Our results showed that differences in ET between AF and MC were not significant. Instead, we found significantly higher evapotranspiration indices during a drier than normal year compared to a wet year across sites and methods. This shows that the potentially small effect from the trees on ET was overlaid by the effect of local climatic

conditions. In addition, we found a similar plant physiological response of the AF and the MC systems, characterized by small differences between canopy resistances.

   Overall, we conclude that the inclusion of tree strips into the agricultural landscape has not resulted in significant water losses to the atmosphere via ET and agroforestry can be a land-use alternative to conventional agriculture.

*Data availability.* All data used for the figures in this manuscript will be provided after acceptance.

*Author contributions.* CM designed and performed the field work, analysed the data and has written the current manuscript. AK and LS wrote the project scientific proposal, acquired the funding as part of the BonaRes SIGNAL consortium, and contributed to field work and analysis. All authors contributed to the discussion and manuscript writing.

*Competing interests.* The authors declare no competing interests.



*Acknowledgements.* We kindly acknowledge the funding from the German Federal Ministry of Education and Research (BMBF, project BonaRes, Modul A: Signal 031A562A) and from the Deutsche Forschungsgemeinschaft (INST 186/1118-1 FUGG). We further wish to acknowledge contributions by M. Herbst to the BonaRes SIGNAL proposal and project design as well as the technical support of field work received by F. Tiedemann, E. Tunsch, D. Fellert, M. Lindenberg, J. Peters (bioclimatology group) and D. Böttger (soil science group of
5   tropical and subtropical ecosystems) from the University of Göttingen.

    This work used eddy covariance data acquired and shared by the FLUXNET community, including these networks: AmeriFlux, AfriFlux, AsiaFlux, CarboAfrica, CarboEuropeIP, CarboItaly, CarboMont, ChinaFlux, Fluxnet-Canada, GreenGrass, ICOS, KoFlux, LBA, NECC, OzFlux-TERN, TCOS-Siberia, and USCCC. The ERA-Interim reanalysis data are provided by ECMWF and processed by LSCE. The FLUXNET eddy covariance data processing and harmonization was carried out by the European Fluxes Database Cluster, AmeriFlux Man-
10   agement Project, and Fluxdata project of FLUXNET, with the support of CDIAC and ICOS Ecosystem Thematic Center, and the OzFlux, ChinaFlux and AsiaFlux offices.



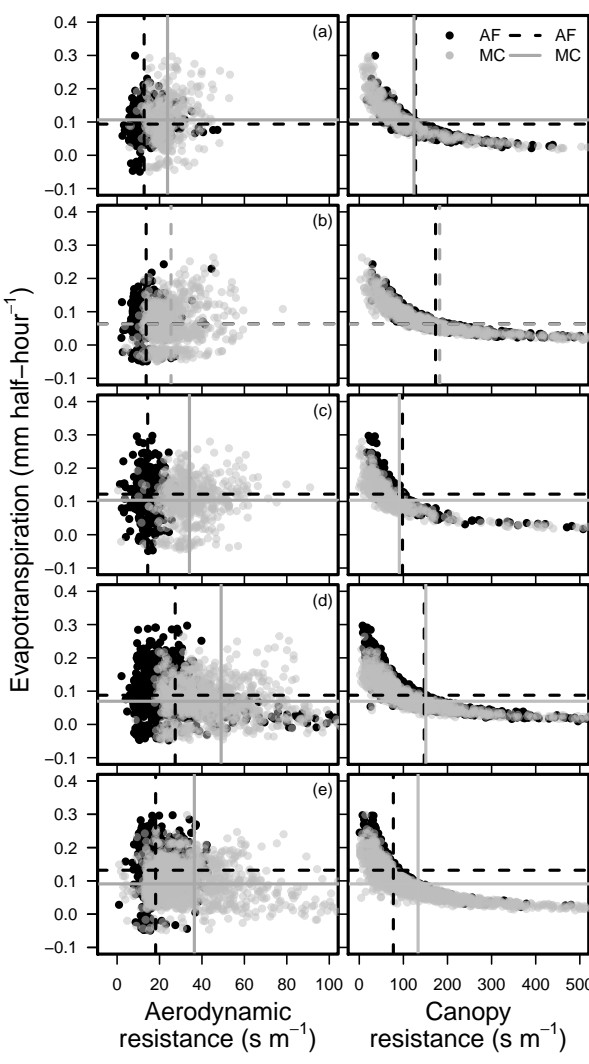

**Figure 11.** Half-hourly evapotranspiration rates from EC-LC versus aerodynamic resistance (left) and canopy resistance (right) for Dornburg, (a), Forst, (b), Mariensee, (c), Reiffenhausen, (d), and Wendhausen, (e). The dashed grey line corresponds to the mean canopy resistance and evapotranspiration at the AF plot and the dashed black line corresponds to the mean canopy resistance and evapotranspiration at the MC plot at the specific site. Only data corresponding to ideal ambient conditions are shown, e.g. a global radiation, $R_G \geq 400\ \mathrm{W\,m^{-2}}$, a wind speed, $u \geq 1\ \mathrm{m\,s^{-1}}$ and a vapour pressure deficit, VPD$=1\pm0.3$ kPa.



## Appendix A: Tables

**Table A1.** Temporal extend of the EC measurement campaigns.

| Site | Campaign period |
|------|-----------------|
| Dornburg MC | 16 June to 14 July 2016 |
| Donburg AF | 14 July to 12 August 2016 |
| Reiffenhausen AF | 12 August to 14 September 2016 |
| Wendhausen | 03 May to 02 June 2017 |
| Forst | 08 June to 08 July 2017 |
| Mariensee | 21 July to 19 September 2017 |

**Table A2.** Mean air temperature, $T_A$, vapor pressure deficit, VPD, global radiation, $R_G$, and the cumulative precipitation, P, for the respective site and campaign period.

| Site | $T_A$ (°C) | P (mm) | VPD (hPa) | $R_G$ (Wm$^{-2}$) |
|------|-----------|--------|-----------|-------------------|
| Dornburg AF | 19.0 | 57.1 | 6.41 | 200.7 |
| Dornburg MC | 18.6 | 2.1 | 7.35 | 212.6 |
| Forst AF | 21.4 | 18.9 | 12.02 | 358.8 |
| Forst MC | 21.2 | 14.8 | 11.88 | 371.5 |
| Mariensee AF | 18.54 | 40.6 | 6.2 | 258.9 |
| Mariensee MC | 16.93 | 163.5 | 4.7 | 172.8 |
| Reiffenhausen AF | 19.31 | 26.3 | 8.02 | 219.1 |
| Wendhausen AF | 16.6 | 48.6 | 5.4 | 235.0 |
| Wendhausen MC | 15.5 | 90.7 | 5.2 | 239.9 |





**Table A3.** Site specific soil characteristics, with the soil texture being representative for the top soil column of 0.3 m. The bulk density is representative for the top soil column of 0.05 m. Data provided by Göbel et al. (2018) and Schmidt et al. (unpublished data).

| Site | Clay content (%) | Sand content (%) | Bulk density ($\mathrm{kg\,m^{-3}}$) |
|------|------|------|------|
| Dornburg AF | 20.5 | 3.75 | 1.22 |
| Dornburg MC | 38 | 10.75 | 1.19 |
| Forst AF | 7 | 60.75 | 1.3 |
| Forst MC | 9.5 | 66.75 | 1.28 |
| Mariensee AF | 11.75 | 48 | – |
| Mariensee MC | 31.67 | 54.33 | 1.28 |
| Reiffenhausen AF | 23.75 | 31.5 | 1.28 |
| Reiffenhausen MC | 22.75 | 49.75 | 1.28 |
| Wendhausen AF | 35 | 18.25 | 1.085 |
| Wendhausen MC | 44.5 | 27 | 0.89 |





**Appendix B: Figures**

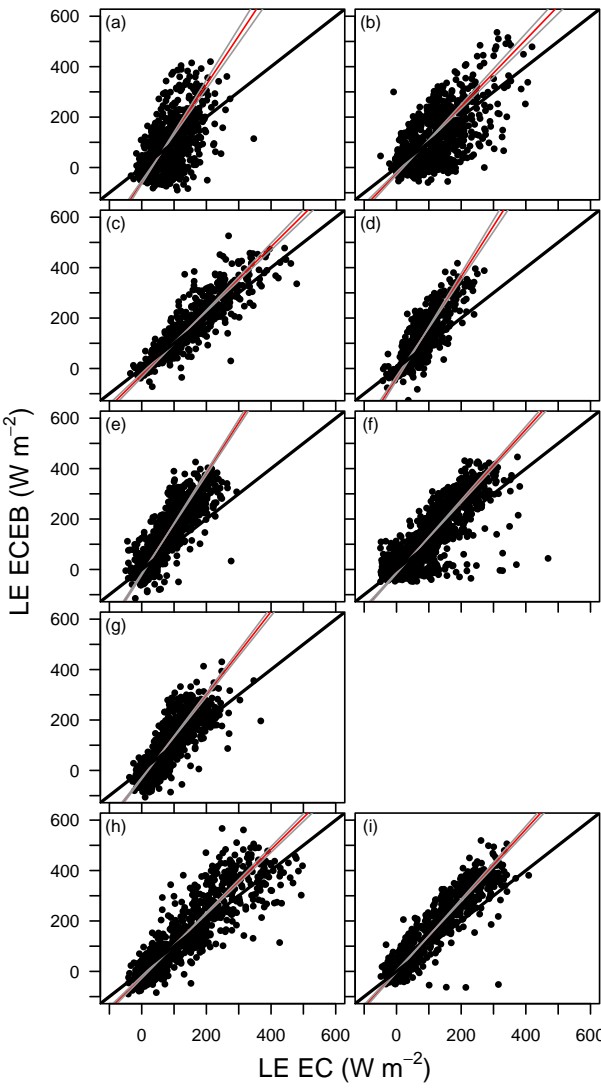

**Figure A1.** Scatter plot of LE from ECEB versus EC at Dornburg agroforestry, (a), Dornburg monoculture, (b), Forst agroforestry, (c), Forst monoculture, (d), Mariensee agroforestry, (e), Mariensee monoculture, (f), Reiffenhausen agroforestry, (g), Wendhausen agroforestry, (h), and Wendhausen monoculture, (i). The red line denotes the best fit line with grey lines the $\pm 2.5\%$ confidence interval lines and the solid black lines corresponds to the 1:1 line.



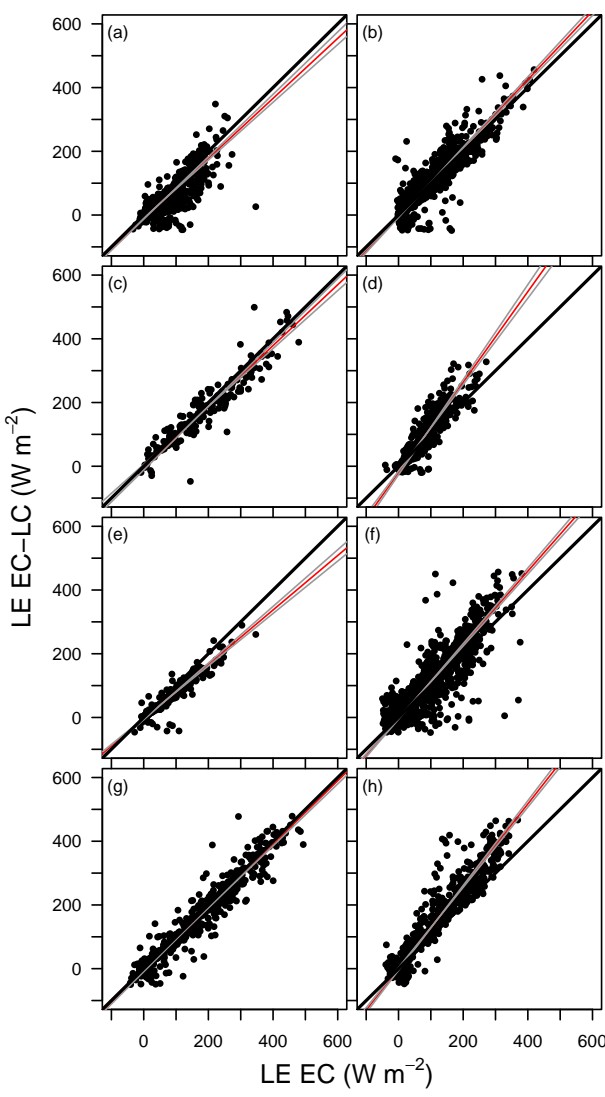

**Figure A2.** Scatter plot of LE from EC-LC versus EC at Dornburg agroforestry, (a), Dornburg monoculture, (b), Forst agroforestry, (c), Forst monoculture, (d), Reiffenhausen agroforestry, (e), Mariensee monoculture, (f), Wendhausen agroforestry, (g), and Wendhausen monoculture, (h). The red line denotes the best fit line with grey lines the ±2.5% confidence interval lines and the solid black lines corresponds to the 1:1 line.
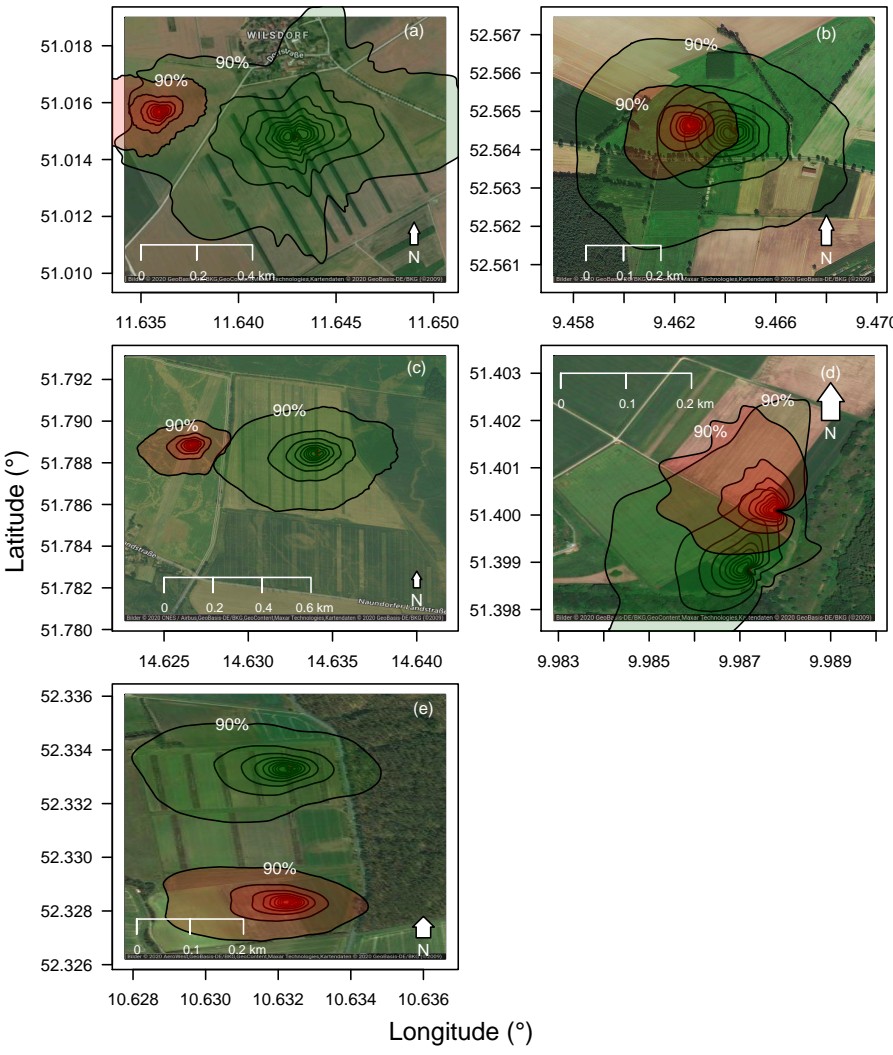

**Figure A3.** Flux footprint climatology for Dornburg, (a), Mariensee, (b), Forst, (c), Reiffenhausen, (d), and Wendhausen, (e), for both all data available during the years 2016 and 2017. Green shaded footprints correspond to the agroforestry plot and red shaded footprints correspond to the monoculture plot. For the analysis only daytime data were used ($R_G > 20$ Wm−2). Aerial photographs originate from Google maps/ Google earth ©Google 2020.



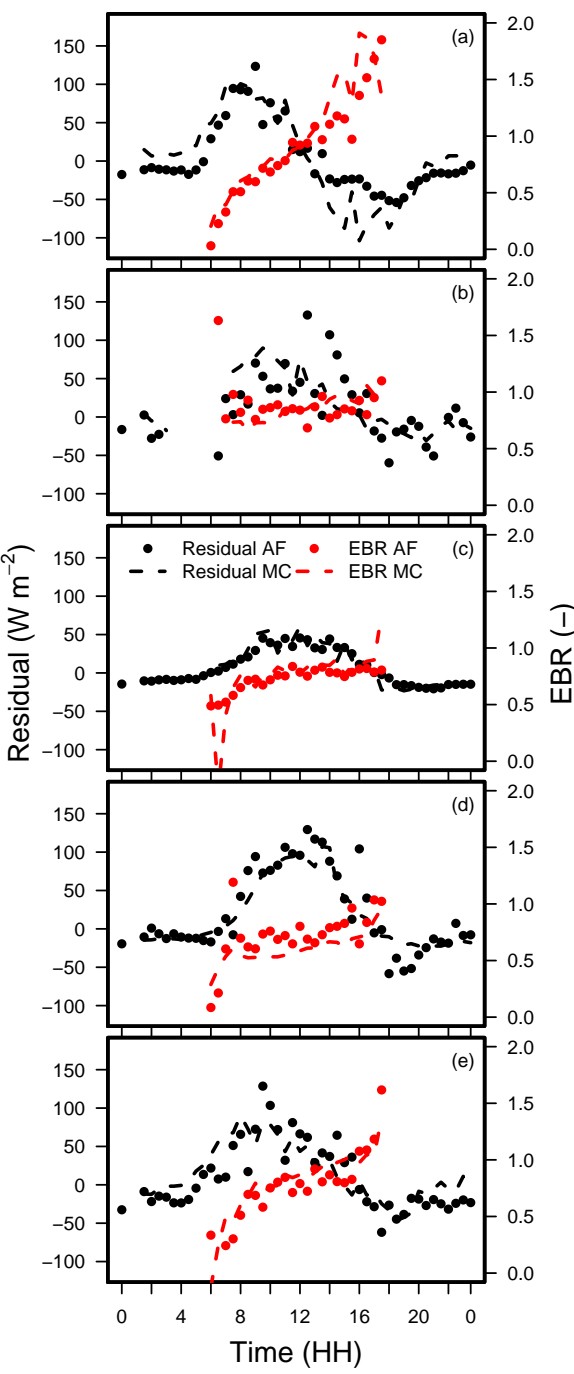

**Figure A4.** Daily median of the energy balance ratio (EBR) and the residual energy at Dornburg, (a), Forst, (b), Mariensee, (c), Reiffenhausen, (d), and Wendhausen, (e), for the agroforestry (AF) and the monocultural plots (MC) separately. The latent and sensible heat flux was calculated following the low-cost eddy covariance set-up.

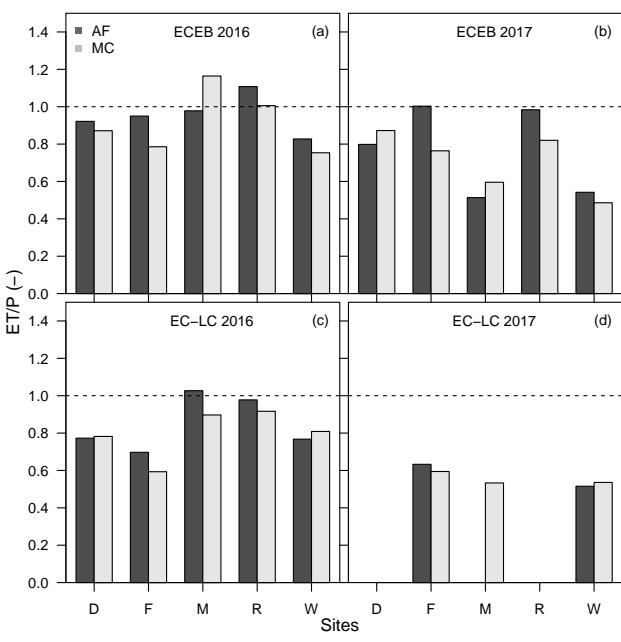

**Figure A5.** Bar plot of the evapotranspiration index for the ECEB method for the years 2016, a, and 2017, b, and for the EC-LC method for 2016, c, and 2017, d, for sites, e.g., Dornburg, "D", Forst, "F", Mariensee, "M", Reiffenhausen, "R", and Wendhausen, "W". The dashed line indicates a evapotranspiration index of one.

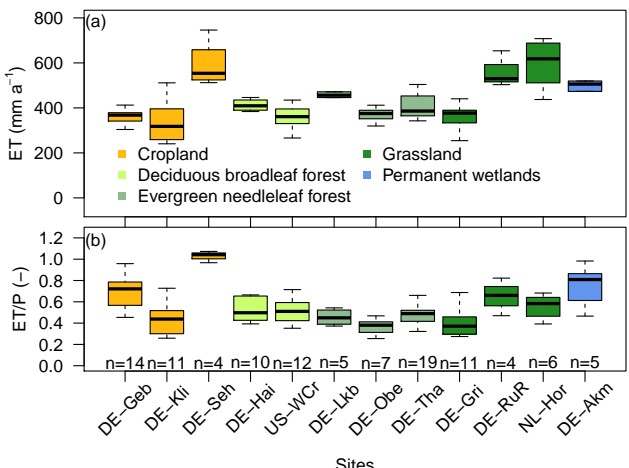

**Figure A6.** Box-plot of the annual summed ET, a, and the evapotranspiration index, b, for 12 FLUXNET sites, e.g., Gebesee, "DE-Geb", Klingenberg, "DE-Kli", Selhausen, "DE-Seh", Hainich, "DE-Hai", Willow Creek, "US-WCr", Lackenberg, "DE-Lkb", Oberbärenburg, "DE-Obe", Tharandt, "DE-Tha", Grillenburg, "DE-Gri", Rollesbroich, "DE-RuR", Horstermeer, "NL-Hor", and Anklam, "DE-Akm". The number of years used for the analysis is written at the bottom of the plot.



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
