# Peer review of "Evapotranspiration over agroforestry sites in Germany"

_Biogeosciences, 2020_

## Referee Comment (RC1) · Anonymous Referee #1 · 2 Jul 2020

This paper presents ET measurements from paired monoculture/agro-forestry sites throughout Germany. The results indicate insignificant differences in ET between the land use types, which appears to be a positive result. The writing is adequate, but I personally feel that the document overemphasizes the statistical comparison between the paired sites to the extent that the important message of the paper is obscured. The content of the paper is fine, but the text needs further refinement.

Comments on paper:

Page 1 line 23: Direct comparison of ET between wet and dry years is not very relevant because the available energy is likely different between the two years.

Page 1 lines 16-26: This is the most important point of the paper. However your

description does not speculate or give guidance as to whether you expect higher ET at the AF or MC locations, no hypothesis.

Page 2 line 8: You note that SRC are comparable to monoculture (forestry) but you don't indicate what aspects are comparable - are you refering to energy partitioning and water use?

Page 2 Intro: Most of your references are relatively recent, you might gain some insights by reviewing earlier work. See references in Cleugh.

Page 3 line 10: The ECEB method is not really limited by closure of the energy budget because this is the default assumption for ECEB. It is, however, limited by the accuracy of your estimates of senible heat flux, net radiation, soil heat flux and change in storage terms.

Page 3 line 20: Why do your partition the residual energy budget between just H and LE and not between H,LE and G - or possibly even Rn

Page 3 line 21: I would suggest being more specific in your hypothesis. Specify short-rotation copice agro-foresty, as your results may not extent to other systems.

Page 5 line 4: How did you know if precipitation data were missing?

Page 5 line 10: Did you use the precipitation data from the AF plots? and if so how did you use them?

Page 7 equ 4: Technically, this conversion gives you units of mg/m2 not mm/30 min. (assuming your lambda value is using milligrams and not the more usual grams. This needs to be explicit to avoid readers from incorrectly applying this equation. (i.e. give units for your variables)

Page 8 line 9-10: This sentence needs to be fixed. Also, it is an assumption that lack of energy budget closure reduces ET. That assumption is not necessarily true.

Page 10 line 2: Your Big-Leaf assumption may be appropriate for the MC sites but less

so for the AF sites, can you address the potential effects.

Page 10 Equ 11: Here and elsewhere in the paper you use 'lambda' as the latent heat of vaporization but in the text you use 'L' . Best to use one or the other, not both.

Page 10 line 9: is 'ppp' a variable, if so it should be shortened to a single character.

Page 10, equ 14: don't use VPD as a variable name, reduce it to a single character (e.g. 'D', or a single character variable with a subscript or superscript (e.g. 'e_D')

Page 11 line 2-3: Did you account for wind direction. The AF site is inherently non-homogeneous, and similar to other row-structured crops may have strong directional dependencies.

Page 11 sec 3.1: This information might be more succinctly incorporated as a table - only referring in text to any atypical conditions.

Page 14 line 24-25: Water vapour concentrations are not a good indicator of spectral response - many other factors come into play.

Page 15 fig 4: Why is there no nocturnal data for some sites?

Page 17 sec 3.4: Instead of using "LE from EC", "LE from EC_LC", "LE from ECEB", might I suggest using subscripts LE_a = LE from EC LE_b = LE from ECEB LE_c = LE from EC_LC It will make reading the paper much easier.

Page 20 line 6-14: This is really interesting. I would cut down on the amount of stats provided and focus on the underlying concepts of what be causing this - which obviously is on scales much bigger than the individual sites

Page 20 line 17-18: This seems inconsistent with your preceding paragraph.

Page 20 sec 3.4.3: Not so sure about the usefulness of this section. As presented it is a simple algebraic exploration assuming linear relationships. In reality, changing one or more the components by +/- 20% may have non-linear effects on the other components, which can not be accurately captured by the your current analysis method.

Page 22 line 15: This is perhaps expected, by definition Rn is the sum of the other components.

Page 22 line 27-30: Is it correct that this is an assumption and you did not measure evaporation and transpiration separately.

Page 24 sec 3.5.2: Even though ET was measured by EC only for campaigns, it might be useful to compare sums of ET by all three methods for those campaign periods.

Page 27 line 3: how do you get a displacement height of 7 m with a canopy height of 5 m?

Page 27 line 7-8,13-14: Is these relationship inherent from the derivation of canopy conductance from ET?

---

## Referee Comment (RC2) · Anonymous Referee #2 · 2 Jul 2020

The authors measured evapotranspiration (ET) over pairs of adjacent agroforestry (AF, tree lines plus crop or grassland) and tree-free reference fields (MC for monoculture, only crop or grassland) at five sites in Germany over up to 2 years with 3 different methods. Plain eddy-covariance (EC) was used during campaigns, roving between sites due to limited gas analyzer availability. An energy balance method (ECEB) yielding ET as residual of EC measurements of the sensible heat flux and the non-turbulent energy balance terms, as well as a low-cost (EC_LC) method introduced elsewhere by the authors were operated continuously over the 2-year period and validated against EC. The paper presents - a comparison of the methods, in particular of the continuous methods versus EC - a detailed analysis of the energy balance closure (EBC) problem for the concerned methods (EC and EC-LC), and - a comparison of ET between AF

and MC, between sites and years, and possible explanations on the result of this comparison, in particular on AF vs. MC The study presents the results of an impressive amount of work, applied according to best practice and including innovative elements, to questions relevant to BG and helpful for land use management decisions. The paper is well written and in terms of content and methods integrity could be published as is. However, the choice of the authors to treat so many dimensions (5 sites with 2 plots each, 3 methods, 3 above research questions) in one paper, makes the clear presentation of methodology and results a particular challenge. In this respect the readability of the mansucript could be improved in several ways. As far as I am concerned, all these improvements are optional; implementing some of them would qualify as minor revision and probably describes my recommendation best.

Overarching comments each concerning several points of the paper at once:

1. Whereas the abstract doesn't specify on the nature of the "monoculture" (MC) and the start of the introduction explicitly (and correctly) states that tree plantations can be monocultures as well, it becomes clear only later (maybe only in section 2.1 if I didn't overlook anything) that MC in this study refers exclusively to the crops/grass without trees (as opposed to a dense tree monoculture without a deliberately cropped understorey, which could be just an as relevant and logical comparison partner). Maybe it would be better to replace MC by something like e.g. NT for no-tree. If not I suggest to clarify earlier what MC in this paper means.

2. The fact that the authors seem to have tried (if I didn't misunderstand) both, down-correcting EBEC results to yield ET estimates with an EBC gap (sect 2.3.1, p8L17) and sometimes up-correcting EC and EC-LC results (Eq. 7-9, table 5), makes it hard to follow the interpretation of the results, particularly in places where the authors try to explain differences between methods / fields with their different EBCs (p14L32 / Sect. 3.3). Ideally it should be stated somewhere clearly that you present all results with (then down-correcting ECEB) or without (then up-correcting EC and EC-LC) an anergy balance closure gap. If then having to do the opposite, or a halfway correction,

is still urgently needed for particular tasks in particular places, such as e.g. gap-filling between ECEB and EC-LC, it should be made clear at these points that this is the only purpose and usage of that "other" correction approach.

3. In many figures an important correspondence between sub-panels (e.g. a-e) and cases (mostly sites, sometimes methods or periods) can only be established through the caption, which is even complicated by the letters changing their meaning with respect to site depending on whether one or two sub-panels are needed per site. I suggest to include the most important differences (e.g. site names) in the subpanels or next to rows or panels of subpanels, such that the figure can better stand alone. In Figure 9 quite suddenly abbreviations for the site names are introcudes which were nowhere used before (but might be useful for the above suggestion). It might also be worth thinking about re-naming the sites by characteristics relevant to the interpretation, e.g. crop vs. grass and/or the ranking of tree density.

4. Textbook knowledge that many others would present not at all or in an appendix is reported in the methods section. This is not necessarily a bad thing (although contributing to the overall length), but currently it is not done consistently. Equation 4 and 5 detail on quite straightforward conversion matters, and equations 12 and 13 on saturation vapour pressure and its slope, but on the other hand section 2.2.3 (p7L26) merely states that "mole fraction was calculated using measurements of relative humidity, air temperature and air pressure...", although this conversion involves at least as many reproduction-relevant decisions (and maybe partly same equation(s)) as the ones mentioned before. Ways out could be e.g. either to drop all these details, or insert an appendix section where such equations are gathered, some of which could then be referred to from multiple points of the paper if necessary.

Further comments on the analysis:

5. p08L19 (Sect 2.3.1): I may be overlooking something, and given how little we know about the source of the EBC problem your solution might be as good or bad as the

more widespread Twine partitioning, but I do not understand why the latter cannot be applied to your data. Mathematically a Bowen-ratio conserving correction is equivalent to correcting both fluxes by the same factor 1/EBR, without any explicit need to know/compute/introduce the Bowen Ratio itself (and even if this was the case there would probably be an analytical or iterative solution to the problem). So if the available H (from EC) is already subject to the closure problem and does not need to be "down-corrected", the only thing left to do is to multiply the residually determined LE with EBR to get the desired estimate of a "non-closing / EC-like" LE.

6. p11L05 (Sect. 2.6 / Equation 18): After an elaborate description of how the Penman-Monteith approach is used to infer conductances, the simpler (humidity-free) Priestly-Taylor approach is introduced for the Budyko analysis, although alternatives consistent with Penman-Monteith (e.g. FAO grass reference ET) exist. Was there a particular reason for this decision? Luckily it probably affects all sites similarly (more similar than in a global study mixing very humid and arid sites) and seems only to be needed in Fig. 10, even there only slightly changing X axis position but not the overall pattern.

Further comments on technical / presentation details:

7. p01L14 (abstract): Consider rewording "superior performance" to make clear that this indicates superior agreement with the widespread EC method. This is not necessarily identical to superior performance in capturing true ET.

8. p01L17 (abstract): There is an ongoing debate whether, how much and how we should continue to base conclusions about differences on significance (e.g. Amrhein et al., Nature 567:305, DOI: 10.1038/d41586-019-00857-9). While reporting p-values in a paper for the sake of completeness cannot do much harm (without wrong interpretation), care should be taken especially where wrong use in the past was particularly popular, and one of these cases is inferring that a difference is nonexistent or unimportant from a "failed" (insignificant) test. This sentence (and versions of it in the main text) comes somewhat close to suggesting something like this (although not explicitly

claiming it). It might be more convincing to show (as done in the main text) how small the difference actually was (maybe compared e.g. to the mean ETs or to the inter-site, inter-period, or inter-method variability that was probably at the bottom of the significance test) and then it could still be mentiond if wanted (here or elsewhere) that the difference was also statistically insignificant (which depends strongly on sample size even if all the means and variances stay the same, and unlike conclusions from significant results, conclusions from insignificant results have the property to become the more likely the smaller the sample size). Also note that if keeping reporting the p-values somewhere, they should be rounded to a reasonable number of digits; especially for the second one at L23, p = 0.0007 or p < 0.001 would be sufficient.

9. p02L08 (Sect. 1): "comparable" reads strange in this context. Basically they are, aren't they? As far as I know the term monoculture does not distonguish between agriculture and forestry. Also see comment 1.

10. p02L32 (Sect. 1): "such as" reads strange in this context. Maybe ", i.e." instead?

11. p03L01 (Sect. 1): depend*s*

12. p03L29 (Sect. 2.1): While reporting the access date of an URL is important if that URL is a source of data/information that couldn't be replaced by a better source, in this case the URL more has the role of an advertisement or a reference to further information for interested readers, and what exactly they will find at the project site if it still exists is not relevant to the paper. For this an access date seems inappropriate. If you weren't asked to add it during the access review, I'd suggest to remove it.

13. p04L01 (Figure 1): Maybe add a scale bar to the aerial views (or one scale information for all if they are the same). I wonder how wide the elongated MC strip at Forst (b) was, how different the management west and east of it was, and how this is reflected in Sect. 3.2 (footprint analysis).

14. p05L1 (Table 1): System size. Specify if it refers to AF, MC or the sum of both.

15. p05L18 (Sect. 2.2.1): Did I understand correctly that this required at least two available Li-7200? If yes clarify, if no reword sentence.

16. p06L01 (Table 2): ppp for pressure seems unusually long/complicated. Also, in the row that is solely about ppp it looks a bit lonely (and hard to understand) without the long explanation "Atmospheric pressure". I acknowledge that you aimed at consistently giving the long name only upon first occurrence in the table, but here would be space and reason enough for an exception. Or maybe the row could be switched with the BME280.

17. p07L19: "unpublished data" and then no matching entry in the reference list is a bit vague. If there is not even an internal report to refer to (which could then be listed in the references), "pers. comm." would probably be more appropriate, and at any rate in this case the institutional affiliation of Schmidt et al. should be given e.g. in the acknowledgement, to ensure traceability.

18. p07L28 (Sect. 2.2.3): Even though referring to a publication about the method where all this can probably be read in detail, not mentioning that there was a (probably large) spectral loss correction falls back behind the information given in the introduction (p03L17), and will make readers looking for this information in the methods section (the most logical place) wonder if and how this method could work at all.

p08L27 (Sect. 2.3.2): This sentence at a first glance seems to contradict the sentence at the top of the same page. Maybe start like this: "Unlike for the methodological comparison and energy balance analysis, a gap-filling of EC-LC could not be avoided for [this and that, surely not for annual ET sums]. Therefore, for these analyses..."

p09L10-15 (Sect. 2.4): At the beginning consider replacing "As the" by "By". Citing software tools / packages can be useful when a) advertising that the own code can be made available to the reader or when b) Reproduction of results depends on using the same tool (mention package, e.g. because the method is so complicated it might give different results in other langauges). The major axis however is a statistical term

independent of and introduced before R, and if correctly implemented in the package should yield the same result as any self-written implementation. Therefore it seems more important to refer to a statistical textbook or paper - e.g. Webster 1997, Eur. J. Soil Sci. 48:557, doi:10.1111/j.1365-2389.1997.tb00222.x (which by the way also provides in its "calibration" section support for your decision in other places to treat variables to be filled as "dependent" (Y) variables in a regression).

p13L01 (Fig. 3): Cannot see MC footprint in subpanel d, is this somehow related to the inavailability of a campaign at Reiffenhausen mentioned at p08L24? But footprint modelling only relies on data measured anyway by the EC-H setup needed for ECEB and EC-LC? Maybe it would be good to state in a prominent place (or each time a particular result seems to be missing, e.g. in Fig. 3d, Fig. 4 between g and h, Table 3 row Mariensee EC-LC, Fig. 5, Table 4, Fig. 9) what was the reason (in most cases it seems to be the missing campaign at Reifenhausen MC, but not so e.g. in table 3).

p17L26 (Sect. 3.4.2 / describing Fig. 6): Is "square" a commonly recognized or self-explaining description of this kind of diel curve?

p20L29 (Sect. 3.4.3): It took (me) several readings to understand how and why you changed magnitudes, after talking about measured data all the time. Basically the idea of this whole section is simpler and more straightforward than it looks, and if needing to shorten the paper, this (writing it simpler or dropping it completely) would be my first suggestion. It can be reduced to the message "the importance of a relative uncertainty in a flux for the EBC scales with the magnitude of that flux". Even this effect probably vanishes when looking at absolute rather than relative errors / uncertainties, and even though it is not completely irrelevant for deciding how much to invest into improving which flux, it could probably also be demonstrated in a more general way with symbolic maths or a thought experiment.

p25L04 (Sect. 3.5.2) "related" reads strange in this context, maybe "plots with an ET index".

p26L05 (Sect. 3.5.3): "reduce" or "reduced"? The former simply repeats (and takes for granted, but this semms save to me) what the cited references state, while the latter implies a claim that it can be seen well in your data, which should then however be confirmed by a clearer statement.

p26L01 (Sect. 3.5.3): The methodology section preferred aerodynamic conductance, here aerodynamic resistance (the inverse) is used. Consistenly using only resistance or conductance could help to avoid confusion.

p31L06 (acknowledgements): Data from other sites than your own seem to have been used only in one place of the appendix, Fig. A6, if I didn't overlook something. If it is needed at all (there seems to be little connection to the main text), the small amount of sites used there seems to suggest that acknowledgements to the individual site PIs is at least as, or more, important than to the (for this number of sites quite lengthy) list of networks.

---

## Author Response (AR1)

**Final response to the reviewers comment from Anonymous Referee #1 on the manuscript bg-2020-171: "Evapotranspiration over agroforestry sites in Germany"**

We thank you for your feedback, suggestions and helpful comments on the manuscript. In the current document we give a point-by-point answer on above referee report.
We show first the referee comments **(RC)** and second the authors answer **(AR)**. Specific changes in the revised manuscript are marked as green text as part of the authors response.

**0. RC:** *This paper presents ET measurements from paired monoculture/agro-forestry sites throughout Germany. The results indicate insignificant differences in ET between the land use types, which appears to be a positive result. The writing is adequate, but I personally feel that the document overemphasizes the statistical comparison between the paired sites to the extent that the important message of the paper is obscured. The content of the paper is fine, but the text needs further refinement.*

**0. AR:** Thank for very much for your positive feedback and the detailed and valuable comments. We reduced discussions on the statistical significance of the differences between ET from the different land-uses both in Abstract as well as in the main text. We are confident that the quality of the manuscript will improve after considering your suggestions.

**1. RC:** *Page 1 line 23: Direct comparison of ET between wet and dry years is not very relevant because the available energy is likely different between the two years.*

**1. AR:** We used this comparison to test if we can detect the effect of a wet and a dry year on ET fluxes with the used methods. With this analysis we showed that both methods (ECEB and EC-LC) were suitable to detect differences in ET due to different ambient conditions. But, we also showed that differences in ET between the two land-uses and between the two methods were of similar magnitude and of the same sign (ET_AF > ET_MC; ET_ECEB>ET_EC-LC). This makes it difficult to decide whether differences in ET between AF and MC are caused by the presence of the trees of the AF system, or if the differences are an effect of the methodological uncertainties. We clarified this in the abstract, as also shown in the **2. AR** below.

The aforementioned discussions refer to Figure 10 of the initially submitted manuscript. We added a second figure next to the existing one and zoomed into the centre of the plot to indicate the trends. In addition, we used the radiative dryness index on the x-axis instead of the ratio between potential ET and precipitation. The different approaches for potential ET resulted in large differences of up to 100 mm a$^{-1}$. See the new figure with the extended caption below:

[Figure]

**Figure 11. (a)** Evapotranspiration index (ET/P) versus the radiative dryness index ($R_n/\lambda P$) for both land-uses (AF: filled triangles and dots; MC: empty triangles and dots), both set-ups (ECEB: dots; EC-LC: triangles) and both years (2016: red; 2017: blue). The bold black line describe regions of an energy limitation ($R_n/\lambda P<1$) and a water limitation ($R_n/\lambda P>1$). The arrows indicate mean trends of ET for the effect of different years (black arrow), different methods (blue arrow) and different land-uses (grey arrow). **(b)** Trends of the mean evapotranspiration index (ET/P) versus the mean radiative dryness index ($R_n/\lambda P$) for the effect of different years (black), different methods (blue) and different land-uses (grey) extracted from figure (a).

**2. RC:** *Page 1 lines 16-26: This is the most important point of the paper. However your description does not speculate or give guidance as to whether you expect higher ET at the AF or MC locations, no hypothesis.*

**2. AR:** → we formulated the main objective of this work in the first paragraph of the abstract (Page 1 line 5-6), which was to assess if AF systems have higher ET compared to monoculture systems

We clarified the hypothesis and the objective in the abstract as: "[...] Therefore we hypothesize that short rotation coppice agroforestry systems have higher water losses to the atmosphere via ET, compared to monoculture agriculture without trees. In order to test the hypothesis the main objective was to measure actual evapotranspiration of five AF systems in Germany and compare those to five monoculture systems in close vicinity to the AF systems.[...]"

In addition, we extended the discussion in the abstract in the particular lines with a more precise conclusion:

"With respect to the annual sums of ET over AF and MC, we observed small differences between the two land-uses. We interpret this as an effect of compensating small-scale differences in ET next to and in between the tree strips for ET measurements on system-scale. Most likely, differences in ET rates next to and in between the tree strips are of the same order of magnitude but of opposite

sign and compensate each other throughout the year. Differences between annual sums of ET from the two methods were of the same order of magnitude as differences between the two land-uses. Compared to the effect of land-use and different methods on ET, we found larger mean evapotranspiration indices ($\sum$ET/$\sum$P) across sites for a drier than normal year (2016) compared to a wet year (2017). This indicates that we were able to detect differences in ET due to different ambient conditions with the applied methods, rather than the potentially small effect of AF on ET.

We conclude that agroforestry has not resulted in an increased water loss to the atmosphere indicating that agroforestry in Germany can be a land-use alternative to monoculture agriculture without trees."

**3. RC:** *Page 2 line 8: You note that SRC are comparable to monoculture (forestry) but you don't indicate what aspects are comparable - are you refering to energy partitioning and water use?*

**3. AR:** we refer to the geometrical structure of those systems, rather than energy partitioning or water use; SRCs are not mixed systems like agroforestry (trees and crops), they consist only of one tree species, which is similar to monoculture systems with only one crop species, we changed it in the text as follows:

"SRC plantations are monoculture systems with a single tree species grown."

**4. RC:** *Page 2 Intro: Most of your references are relatively recent, you might gain some insights by reviewing earlier work. See references in Cleugh.*

**4. AR:** We looked through older literature, in particular the review from *Cleugh 1998*, and added the idea of a potential increase of evaporation/transpiration in the quiet zone, when dry air advection is missing, the soil water supply limited and the wind velocity reduced. This process again refers to the area next to the tree strips and how those small-scale effects can be referred to system-scale ET remains unclear.

**5. RC:** *Page 3 line 10: The ECEB method is not really limited by closure of the energy budget because this is the default assumption for ECEB. It is, however, limited by the accuracy of your estimates of senible heat flux, net radiation, soil heat flux and change in storage terms.*

**5. AC:** indeed, we changed this as follows:

"The ECEB method is limited by the accuracy of the energy balance components (the net radiation, the sensible heat flux, the ground heat flux and storage terms), typically leading to an overestimation of latent heat fluxes."

**6. RC:** *Page 3 line 20: Why do your partition the residual energy budget between just H and LE and not between H,LE and G - or possibly even Rn*

**6. AC:** Despite substantial research into the partitioning of the energy balance residual (*i.e. Mauder et al. (2017): 'Evaluation of energy balance closure adjustment methods by independent evapotranspiration estimates from lysimeters and hydrological simulations'*) there is no general consensus how to partition the energy balance residual, and the partitioning is likely site and case specific and would require additional information beyond the typical set of measurements we had available. However, research (*Foken et al. (2008): Micrometeorology*) seems to suggest that the largest fraction of the residual is related to the turbulent fluxes (H + LE), rather than to the measurement of available energy. Therefore, we partition the residual only to LE and H.

**7. RC:** *Page 3 line 21: I would suggest being more specific in your hypothesis. Specify short-rotation copice agro-foresty, as your results may not extent to other systems.*

**7. AC:** we changed the hypothesis in the introduction and added the hypothesis to the Abstract as well as a related objective:

"The main hypothesis of the current work was that short-rotation coppice agroforestry systems have higher water losses to the atmosphere via ET, compared to monoculture agriculture without trees."

**8. RC:** *Page 5 line 4: How did you know if precipitation data were missing?*

**8. AC:** we sampled the meteorological data every 10 seconds; for our analysis we checked how many 10 second values per day were available and compared those to the theoretical number → 10 sec values available/10 sec values theoretical

**9. RC:** *Page 5 line 10: Did you use the precipitation data from the AF plots? and if so how did you use them?*

**9. AC:** we did not use the precipitation data from the AF, as the data were strongly affected by interception and not really representative for the AF system, as the precipitation in between the tree strips is expected to be higher than within the tree strips, we used only precipitation from the monoculture system; the annual sums in precipitation between AF and MC differed substantially (AF<<MC), which would have affected the ratios between ET and P. We used only precipitation from the MC sites under the assumption that the mean annual sum of precipitation between AF and MC do not differ due to the small size of the agroforestry systems and no small scale effects on precipitation formation. We added further explanations in the text:

"[...]Therefore, we used the precipitation measurements from the MC system to compute ratios of annually summed actual and potential ET to precipitation at both AF and MC systems. We assume that the annual sum of precipitation at the AF and the MC systems do not differ, due to the relatively small size of the agroforestry systems and no expected local effects of the agroforestry systems on the precipitation formation.[...]"

**10. RC:** *Page 7 equ 4: Technically, this conversion gives you units of mg/m2 not mm/30 min. (assuming your lambda value is using milligrams and not the more usual grams. This needs to be explicit to avoid readers from incorrectly applying this equation. (i.e. give units for your variables)*

**10. AC:** we corrected the formula as shown below:

> Half-hourly evapotranspiration rates in units of $\mathrm{mm\,30\,min^{-1}}$ were calculated from LE as
>
> $$ET = \frac{LE_{ECEB}(\mathrm{J\,kg^{-1}\,s^{-1}})}{L(\mathrm{J\,kg^{-1}})} \cdot 1800(\mathrm{s\,30min^{-1}}) \cdot \frac{1}{\rho_{H_2O}}(\mathrm{m^3\,kg^{-1}}) \cdot 1000\,\mathrm{mm\,m^{-1}} \tag{4}$$
>
> with L $(\mathrm{J\,kg^{-1}})$ the latent heat of vaporization (Dake, 1972) depending on air temperature T $(\mathrm{^{\circ}C})$
>
> $$L = (2.501 - 0.00237\,T) \cdot 10^6, \tag{5}$$
>
> and $\rho_{\mathrm{H_2O}} = 1000\,\mathrm{kg\,m^{-3}}$ the density of liquid water.

**11. RC:** *Page 8 line 9-10: This sentence needs to be fixed. Also, it is an assumption that lack of energy budget closure reduces ET. That assumption is not necessarily true.*

**11. AC:** we reformulated the sentence:

"[...] We corrected \unit{ET_{ECEB}} for the average energy balance non-closure, which we estimated from direct LE measurements by EC during measurement campaigns of minimum four weeks duration. In the current study we found that considering the energy balance residual reduces \unit{ET_{ECEB}}. [...] "

**12. RC:** *Page 10 line 2: Your Big-Leaf assumption may be appropriate for the MC sites but less so for the AF sites, can you address the potential effects.*

**12. AC:** The big-leaf assumption might be violated over AF due to the heterogeneity of the system, this could potentially be a problem. In this discussion heterogeneity refers to the different plant species (crops/grasses and trees) of different heights. The trees infer a shaded area in terms of wind and incident radiation in the quiet zone. On the one hand the reduction in incident radiation might lead to reduced ET due to a different leaf stomata regulation from sunlit and shaded leafs both from trees and crops as well as due to reduced wind velocities. On the other hand trees and crops in the windward site are affected by increased wind velocities and varying incident radiation. So, the big-leaf assumption might even be valid over agroforestry systems due to the compensation of the effects in the lee and at the windward site of the tree strips.
Therefore, the canopy resistance derived from meteorological measurements at our flux tower (one flux tower over AF and MC, respectively) might still be representative for the agroforestry system, as the mean meteorological conditions are recorded. We added some more explanations+discussions in the Mats+Methods section 2.6 ("Canopy resistance") of the revised manuscript:

"Effects of structural differences between AF and MC on ET were studied in terms of the relationship between the aerodynamic and canopy resistances (\unit{s\,m^{-1}}) and half-hourly ET. The canopy resistance was calculated from the rearranged Penman-Monteith equation (Eq. (\ref{equ:PM_equ})) for evapotranspiration, which depends on the canopy conductance, $g_c$

(\unit{m\, s^{-1}}), and the aerodynamic conductance for heat, $g_{ah}$ (\unit{m\, s^{-1}}). The canopy conductance follows the big leaf assumption, assuming that the whole canopy response to environmental changes equals the response of a single leaf. This assumption is valid for the monoculture system with a single crop type of similar height. For the agroforestry systems this assumption is violated due to the different plant species (trees and crops) of different heights. In the lee of the tree strips the reduced wind speed and incident radiation might lead to reduced ET due to a different leaf stomata regulation of sunlit and shaded leafs. In the windward site of the tree strips trees and crops are affected by increased wind velocities and varying incident radiation, thus opposite conditions compared to the lee of the tree strips. However, we assume that the meteorological data from our flux tower represent the mean state of the meteorological conditions within the agroforestry system. Therefore, we are confident that the big-leaf assumption is also valid for agroforestry systems."

**13. RC:** *Page 10 Equ 11: Here and elsewhere in the paper you use 'lambda' as the latent heat of vaporization but in the text you use 'L'. Best to use one or the other, not both.*

**13. AC:** we changed it from 'lambda' to L

**14. RC:** *Page 10 line 9: is 'ppp' a variable, if so it should be shortened to a single character.*

**14. AC:** we changed it from ppp to $P_A$

**15. RC:** *Page 10, equ 14: don't use VPD as a variable name, reduce it to a single character (e.g. 'D', or a single character variable with a subscript or superscript (e.g. 'e_D')*

**15. AC:** we changed VPD to D

**16. RC:** *Page 11 line 2-3: Did you account for wind direction. The AF site is inherently non-homogeneous, and similar to other row-structured crops may have strong directional dependencies.*

**16. AC:** no, we neither account for any wind direction, nor will we include a new analysis

**17. RC:** *Page 11 sec 3.1: This information might be more succinctly incorporated as a table - only referring in text to any atypical conditions.*

**17. AC:** we kept this section and shortened it; we added Table A2 to this section as follows:

**3.1 Meteorological conditions and plant physiological stages during the campaigns**

For the meteorological conditions during the campaigns we refer to time series of relevant meteorological parameter in Figure 2 and mean values in Table 3.

**Table 3.** Mean air temperature, $T_A$, vapor pressure deficit, D, global radiation, $R_G$, and the cumulative precipitation, P, for the respective site and campaign period.

| Site | $T_A$ (°C) | P (mm) | D (hPa) | $R_G$ (Wm$^{-2}$) |
|---|---|---|---|---|
| Dornburg AF | 19.0 | 57.1 | 6.41 | 200.7 |
| Dornburg MC | 18.6 | 2.1 | 7.35 | 212.6 |
| Forst AF | 21.4 | 18.9 | 12.02 | 358.8 |
| Forst MC | 21.2 | 14.8 | 11.88 | 371.5 |
| Mariensee AF | 18.54 | 40.6 | 6.2 | 258.9 |
| Mariensee MC | 16.93 | 163.5 | 4.7 | 172.8 |
| Reiffenhausen AF | 19.31 | 26.3 | 8.02 | 219.1 |
| Wendhausen AF | 16.6 | 48.6 | 5.4 | 235.0 |
| Wendhausen MC | 15.5 | 90.7 | 5.2 | 239.9 |

**18. RC:** *Page 14 line 24-25: Water vapour concentrations are not a good indicator of spectral response - many other factors come into play.*

**18. AC:** We changed the text to reflect that the spectral response characteristics of the two analyser were similar as follows:

"[...] fluctuations were attenuated. The spectral response characteristics of the gas analyser and the thermohygrometer set-up were similar. Therefore, the correction of high-frequency losses is expected to be higher for the compromised gas analyser at the respective MC systems, than for a fully functional gas analyser."

**19. RC:** *Page 15 fig 4: Why is there no nocturnal data for some sites?*

**19. AC:** There was not enough power available to cover the power needs, due to the solar power supply of the station. Therefore, we had larger data losses during night. We included an explanation to the figure:

"Gaps in nocturnal data are due to the limited power availability from the solar power supply."

**20. RC:** *Page 17 sec 3.4: Instead of using "LE from EC", "LE from EC_LC", "LE from ECEB", might I suggest using subscripts LE_a = LE from EC LE_b = LE from ECEB LE_c = LE from EC_LC It will make reading the paper much easier.*

**20. AC:** it is a good suggestion, we changed the text as follows

$LE_{ECEB}$, $LE_{EC}$ and $LE_{EC\text{-}LC}$ for LE and
$ET_{ECEB}$, $ET_{EC}$ and $ET_{EC\text{-}LC}$ for ET

**21. RC:** *Page 20 line 6-14: This is really interesting. I would cut down on the amount of stats provided and focus on the underlying concepts of what be causing this - which obviously is on scales much bigger than the individual sites*

**21. AC:** Indeed, we discussed partly in the manuscript that circulations bigger than the individual sites might cause the observed pattern. If this is really the case, we would require additional information beyond the typical set of measurements we had available. Therefore, we did as suggested and cut down the amount of statistics.

"[...]Interestingly, the diel pattern of the EBR from \unit{LE_{EC}} at both land-uses at all sites are equal. Additionally, the differences between the median diel cycle EBRs (between 6 am and 6 pm) at the AF and the MC system were small, with differences of minimum -0.09 and maximum 0.13 across sites. As both flux towers located at the AF and the MC system at one site are separated by approximately 100 to 500 m and the diel patterns look similar, we suspect that the non-closed surface energy balance at one site is caused by local effects of longer wavelength than the commonly applied averaging period of 30 minutes and beyond the individual site level.[...]"

Additionally, we added a figure with median diurnal cycles of the different energy balance components for the Dornburg AF and MC sites to explain the unexpected diel cycle of the energy balance ratio at the sites. We added some more text to strengthen our interpretation of an advection effect on the observed energy balance ratio.

Here the new figure:

[Figure]

**Figure 7.** Median diurnal cycle of the energy balance components for Dornburg AF and MC for the campaign times (Table A1).

And here the extended text:

> The Dornburg site might be affected by horizontal advection of moisture and heat. Oncley et al. (2007) reported that the advection of moisture had the highest contribution to the unclosed energy balance compared to the other components and the maximum peak of the horizontal moisture advection term was in the afternoon, as energy was accumulated during the day and released in the afternoon. We suspect that this is also the case for the Dornburg site. The sensible heat flux follows the diurnal cycle of available energy with the maximum peak at midday at the agroforestry and the monoculture system (Fig. 7). In contrast, the median of the latent heat flux had its maximum in the afternoon at around 2 pm and was positive even after the available energy changed its sign.
>
> In addition to advective transport, the unclosed surface energy balance could be related to energy storage terms such as biomass, the air or photosynthesis (Jacobs et al., 2008), that have previously not been considered. The pattern seen at Dornburg may be attributed to a release of energy during the afternoon, which correspond to a surplus of energy and a better closure of the energy balance. In the morning hours the storage terms have an opposite sign, which correspond to a lack of energy and a subsequent poorer energy balance closure. Considering the storage terms would lead to a reduction of the residual energy and a better closure of the energy balance.

**22. RC:** *Page 20 line 17-18: This seems inconsistent with your preceding paragraph.*

**22. AC:** indeed, we removed the discussion on the residual energy, as this is causing the diel cycle of the energy balance ratio, which we explained

**23. RC:** *Page 20 sec 3.4.3: Not so sure about the usefulness of this section. As presented it is a simple algebraic exploration assuming linear relationships. In reality, changing one or more the components by +/- 20% may have non-linear effects on the other components, which can not be accurately captured by the your current analysis method.*

**23. AC:** We removed this section from the paper

**24. RC:** *Page 22 line 15: This is perhaps expected, by definition Rn is the sum of the other components.*

**24. AC:** Yes, this is correct. We removed this section from the paper.

**25. RC:** *Page 22 line 27-30: Is it correct that this is an assumption and you did not measure evaporation and transpiration separately.*

**25. AC:** yes, this is an assumption and we changed it accordingly:

"We assume that after the ripening of the crops evaporation contributed the most to the measured ET at the MC plot, whereas at the AF plot both evaporation from the crop fields between the tree strips and transpiration from the trees contributed to the measured flux."

**26. RC:** *Page 24 sec 3.5.2: Even though ET was measured by EC only for campaigns, it might be useful to compare sums of ET by all three methods for those campaign periods.*

**26. AC:** we included a new sub-section 3.5.1 'Sums of evapotranspiration during the campaigns' and a figure:

**3.5.1 Sums of evapotranspiration during the campaigns**

Sums of evapotranspiration for all three methods, all sites and the campaign periods indicate higher sums of $ET_{ECEB}$ relative to $ET_{EC}$, except for Dornburg AF (Fig. 8). The difference between sums of $ET_{ECEB}$ and $ET_{EC}$ reflect the unaccounted correction of $ET_{EC}$ and $ET_{ECEB}$ for the energy balance non-closure. The large difference between sums of

$ET_{ECEB}$ and $ET_{EC}$ at Mariensee AF correspond to the low energy balance closure of 65% at the site. Differences between sums of $ET_{EC-LC}$ and $ET_{EC}$ correspond to lower $ET_{EC-LC}$ than $ET_{EC}$ over the AF systems and higher $ET_{EC-LC}$ than $ET_{EC}$ over the MC systems. This is indicated by slopes smaller and higher one of a linear regression analysis between $ET_{EC-LC}$ and $ET_{EC}$ (Table 4).

[Figure]

**Figure 8.** Sums of uncorrected and not gap-filled half-hourly evapotranspiration for all three methods and all sites during the campaign periods. Sites are abbreviated by their first letter and contain either AF for agroforestry or MC for monoculture. Incomplete records with either $ET_{EC}$, $ET_{ECEB}$ or $ET_{EC-LC}$ missing were omitted. Data for $ET_{EC-LC}$ at Mariensee AF are missing due to technical problems of the sensor during the campaign and all data for Reiffenhausen MC are missing due to the unavailability of a campaign.

**27. RC:** *Page 27 line 3: how do you get a displacement height of 7 m with a canopy height of 5 m?*

**27. AC:** yes, this is a typo and we changed it in the text to:

"[...] a displacement height d of 0.7 m and 3.5 m for canopy heights of 1 m and 5 m, respectively."

**28. RC:** *Page 27 line 7-8,13-14: Is these relationship inherent from the derivation of canopy conductance from ET?*

**28. AC:** The canopy resistance was derived as the inverse of the canopy conductance with $ET_{EC-LC}$. Small differences in canopy resistance between the two land-uses are an artefact from small differences in ET between the two land-uses. We did change the title of the derivation of r_c and r_ah from 'Canopy conductance' to 'Canopy resistance' to make this more clear. Additionally, we moved the whole derivation of the canopy resistance and other formulas to the Appendix to keep the overall length of the manuscript short.

**Final response to the reviewers comment from Anonymous Referee #2 on the manuscript bg-2020-171: "Evapotranspiration over agroforestry sites in Germany"**

We thank you for your detailed feedback, suggestions and helpful comments on the manuscript. In the current document we give a point-by-point answer on above referee report. We show first the referee comments **(RC)** and second the authors answer **(AR)**. Specific changes in the revised manuscript are marked as green text as part of the authors response.

**1. RC:** *The authors measured evapotranspiration (ET) over pairs of adjacent agroforestry (AF, tree lines plus crop or grassland) and tree-free reference fields (MC for monoculture, only crop or grassland) at five sites in Germany over up to 2 years with 3 different methods. Plain eddy-covariance (EC) was used during campaigns, roving between sites due to limited gas analyzer availability. An energy balance method (ECEB) yielding ET as residual of EC measurements of the sensible heat flux and the non-turbulent energy balance terms, as well as a low-cost (EC_LC) method introduced elsewhere by the authors were operated continuously over the 2-year period and validated against EC. The paper presents - a comparison of the methods, in particular of the continuous methods versus EC - a detailed analysis of the energy balance closure (EBC) problem for the concerned methods (EC and EC-LC), and - a comparison of ET between AF and MC, between sites and years, and possible explanations on the result of this comparison, in particular on AF vs. MC The study presents the results of an impressive amount of work, applied according to best practice and including innovative elements, to questions relevant to BG and helpful for land use management decisions. The paper is well written and in terms of content and methods integrity could be published as is. However, the choice of the authors to treat so many dimensions (5 sites with 2 plots each, 3 methods, 3 above research questions) in one paper, makes the clear presentation of methodology and results a particular challenge. In this respect the readability of the mansucript could be improved in several ways. As far as I am concerned, all these improvements are optional; implementing some of them would qualify as minor revision and probably describes my recommendation best.*

**1. AR:** Thank for very much for your positive feedback and the detailed and valuable comments. We are confident that the quality of the manuscript will improve after considering your suggestions.

**2. RC:** *1. Whereas the abstract doesn't specify on the nature of the "monoculture" (MC) and the start of the introduction explicitly (and correctly) states that tree plantations can be monocultures as well, it becomes clear only later (maybe only in section 2.1 if I didn't overlook anything) that MC in this study refers exclusively to the crops/grass without trees (as opposed to a dense tree monoculture without a deliberately cropped understorey, which could be just an as relevant and logical comparison partner). Maybe it would be better to replace MC by something like e.g. NT for no-tree. If not I suggest to clarify earlier what MC in this paper means.*

**2. AR:** We prefer the current abbreviation of AF and MC. We clarified in the abstract and the main text what exactly the two abbreviations refer to.

In the Abstract:
"Therefore we hypothesize that short-rotation coppice agroforestry systems have higher water losses to the atmosphere via ET, compared to monoculture agriculture without trees (MC)."

In the introduction:
"The cultivation of fast growing trees with annual crops or perennial grass-lands on the same piece of land is an example of agroforestry (AF) \citep{Morhart2014,Smith2013} and has numerous environmental benefits relative to monoculture (MC) systems consisting only of crops or grasses without trees \citep{Quinkenstein2009}."

**3. RC:** *2. The fact that the authors seem to have tried (if I didn't misunderstand) both, down-correcting EBEC results to yield ET estimates with an EBC gap (sect 2.3.1, p8L17) and sometimes up-correcting EC and EC-LC results (Eq. 7-9, table 5), makes it hard to follow the interpretation of the results, particularly in places where the authors try to explain differences between methods / fields with their different EBCs (p14L32 / Sect. 3.3). Ideally it should be stated somewhere clearly that you present all results with (then down-correcting ECEB) or without (then up-correcting EC and EC-LC) an anergy balance closure gap. If then having to do the opposite, or a halfway correction, is still urgently needed for particular tasks in particular places, such as e.g. gap-filling between ECEB and EC-LC, it should be made clear at these points that this is the only purpose and usage of that "other" correction approach.*

**3. AR:** We treated the data in two different ways:
→ 1. we neither corrected the data up or down for the methodological comparison of the different methods based on the campaigns to explain potential differences between methods, as well as for the energy balance closure estimation (p15 Fig.4, p16 Table 3, p18 Fig. 5, p19 Table 4, p21 Fig. 6 of the initially submitted manuscript)
→ 2. for the comparison of annual sums of ET we did the 'half-way correction' of $ET_{ECEB}$ (down-corrected, both for gap-filling of $ET_{EC-LC}$ and $ET_{ECEB}$) and $ET_{EC-LC}$ (up-corrected) to get closer to reality (as explained in Section 2.3 of the initially submitted manuscript) (p24 Fig. 8, p26 Fig. 9, p27 Tab. 5, p28 Fig. 10 of the initially submitted manuscript )

We included a short explanation on the different gap-filling and energy balance closure adjustment procedures in Section 2.3 of the revised manuscript ("Gap-filling and energy balance closure adjustment"), as shown below:

**2.3 Gap-filling and energy balance closure adjustment**

For the comparison of $ET_{EC}$, $ET_{ECEB}$ and $ET_{EC-LC}$ and the estimation of the energy balance closure during the campaigns, we neither gap-filled the data, nor corrected the data for the energy balance non-closure. For the calculation of annual sums of $ET_{ECEB}$ and $ET_{EC-LC}$ data gaps were filled, and corrected for the energy balance non-closure by distributing the residual equally to H and LE. The residual was estimated by machine learning for times when no data were available. In the following subsections we describe the gap-filling and energy balance closure adjustment procedures for the ECEB and EC-LC set-ups in more detail.

We removed the uncorrected annual sums in Table 5 (initially submitted manuscript) as shown below in the new table (now table No 6 in the revised version of the manuscript), as we never used the information of the uncorrected annual sums of ET_ECEB in the text:

**Table 6.** Annual sums of energy balance closure corrected actual evapotranspiration, ET $(\mathrm{mm\,a^{-1}})$, and precipitation, P $(\mathrm{mm\,a^{-1}})$ for all sites, both set-ups (ECEB and EC-LC) and both years (2016 from April to December, and 2017 from January to December). The annual sums of $ET_{ECEB}$ and precipitation at Reiffenhausen AF and MC in 2017 contain data from 01 January 2017 to 01 July 2017 due to destruction of the station. Annual sums of $ET_{EC-LC}$ for Dornburg AF and MC, Mariensee AF, Reiffenhausen AF and MC in 2017 are missing due to instrument malfunctions.

| Method | ECEB | | EC-LC | | | |
| Sites | ET 2016 | ET 2017 | ET 2016 | ET 2017 | P 2016 | P 2017 |
| --- | --- | --- | --- | --- | --- | --- |
| Dornburg AF | 383 | 500 | 321 | – | 414 | 626 |
| Dornburg MC | 362 | 546 | 325 | – | 414 | 626 |
| Forst AF | 494 | 540 | 363 | 340 | 520 | 538 |
| Forst MC | 409 | 411 | 309 | 320 | 520 | 538 |
| Mariensee AF | 386 | 389 | 405 | – | 394 | 757 |
| Mariensee MC | 459 | 451 | 354 | 404 | 394 | 757 |
| Reiffenhausen AF | 406 | 252 | 358 | – | 366 | 256 |
| Reiffenhausen MC | 368 | 210 | 336 | – | 366 | 256 |
| Wendhausen AF | 410 | 446 | 380 | 424 | 496 | 822 |
| Wendhausen MC | 373 | 400 | 401 | 440 | 496 | 822 |

**4. RC:** *3. In many figures an important correspondence between sub-panels (e.g. a-e) and cases (mostly sites, sometimes methods or periods) can only be established through the caption, which is even complicated by the letters changing their meaning with respect to site depending on whether one or two sub-panels are needed per site. I suggest to include the most important differences (e.g. site names) in the subpanels or next to rows or panels of subpanels, such that the figure can better stand alone. In Figure 9 quite suddenly abbreviations for the site names are introcudes which were nowhere used before (but might be useful for the above suggestion). It might also be worth thinking about re-naming the sites by characteristics relevant to the interpretation, e.g. crop vs. grass and/or the ranking of tree density.*

**4. AR:** we did as suggested and included following abbreviations in following figures in the revised version of the submitted manuscript:
*Figure 1:* instead of (a), (c), (e), (g), (i) we wrote Dornburg, Forst, Mariensee, Reiffenhausen, Wendhausen; we remove the letters from the aerial photograph
*Figure 2:* we replaced (a) by Dornburg MC, (b) by Dornburg AF, (c) by Reiffenhausen AF, (d) by Wendhausen, (e) by Forst, and (f) by Mariensee
*Figures 3 and A3:* instead of (a)-(e) we wrote Dornburg, Mariensee, Forst, Reiffenhausen and Wendhausen
*Figure 4:* instead of the letters (a)-(i) we wrote Dornburg AF, Dornburg MC, Forst AF, Forst MC, Mariensee AF, Mariensee MC, Reiffenhausen AF, Wendhausen AF and Wendhausen MC
*Figure 5:* instead of the letters (a)-(i) we wrote Dornburg AF, Dornburg MC, Forst AF, Forst MC, Mariensee AF, Mariensee MC, Reiffenhausen AF, Wendhausen AF and Wendhausen MC
*Figure 6:* instead of (a)-(e) we wrote Dornburg, Forst, Mariensee, Reiffenhausen and Wendhausen
*Figure 7:* Dornburg AF and Dornburg MC
*Figure 8:* we wrote D-AF, D-MC, F-AF, F-MC, M-AF, M-MC, R-AF, W-AF and W-MC
*Figure 9:* instead of (a)-(e) we wrote Dornburg, Forst, Mariensee, Reiffenhausen and Wendhausen
*Figure 12:* instead of (a)-(e) we wrote Dornburg, Forst, Mariensee, Reiffenhausen and Wendhausen
*Figure A1:* instead of the letters (a)-(i) we wrote Dornburg AF, Dornburg MC, Forst AF, Forst MC, Mariensee AF, Mariensee MC, Reiffenhausen AF, Wendhausen AF and Wendhausen MC
*Figure A2:* instead of the letters (a)-(h) we wrote  Dornburg AF, Dornburg MC, Forst AF, Forst MC, Mariensee MC, Reiffenhausen AF, Wendhausen AF and Wendhausen MC
*Figure A4:*  instead of (a)-(e) we wrote Dornburg, Forst, Mariensee, Reiffenhausen and Wendhausen

**5. RC:** *4. Textbook knowledge that many others would present not at all or in an appendix is reported in the methods section. This is not necessarily a bad thing (although contributing to the overall length), but currently it is not done consistently. Equation 4 and 5 detail on quite straightforward conversion matters, and equations 12 and 13 on saturation vapour pressure and its slope, but on the other hand section 2.2.3 (p7L26) merely states that "mole fraction was calculated using measurements of relative humidity, air temperature and air pressure...", although this conversion involves at least as many reproduction-relevant decisions (and maybe partly same equation(s)) as the ones mentioned before. Ways out could be e.g. either to drop all these details, or insert an appendix section where such equations are gathered, some of which could then be referred to from multiple points of the paper if necessary.*

**5. AR:** To keep the main text concise, and given that some of the equations were already described in Markwitz and Siebicke (2019), presenting the EC-LC set-up, we moved Eqs. 4-6 and Eqs. 11-18 to the appendix and included conversion formulas from section 2.2.3. The Appendix is now structured as follows and can be seen in the revised version of the manuscript.

**6. RC:** *Further comments on the analysis:*
*5. p08L19 (Sect 2.3.1): I may be overlooking something, and given how little we know*
*about the source of the EBC problem your solution might be as good or bad as the more*
*widespread Twine partitioning, but I do not understand why the latter cannot be*
*applied to your data. Mathematically a Bowen-ratio conserving correction is equiva-*
*lent to correcting both fluxes by the same factor 1/EBR, without any explicit need to*
*know/compute/introduce the Bowen Ratio itself (and even if this was the case there*
*would probably be an analytical or iterative solution to the problem). So if the available*
*H (from EC) is already subject to the closure problem and does not need to be "down-*
*corrected", the only thing left to do is to multiply the residually determined LE with EBR*
*to get the desired estimate of a "non-closing / EC-like" LE.*

**6. AR:** The suggested solution of multiplying LE_ECEB with the EBR is somehow limited by the
fact that the EBR from EC was only available for the duration of the measuring campaigns and this
would require the prediction of the EBR. Another solution would be to multiply the mean EBR
(derived as the slope between H+LE and Rn-G from the campaigns) with the 30-min LE_ECEB.
This would be even less accurate due to missing the temporal variability of the EBR throughout the
day and the year. As already stated, the main question is not which method to use, it is rather the
question how the residual gets partitioned to the different components. From our point of view, the
current solution was the only viable option given the current data.

**7. RC:** *6. p11L05 (Sect. 2.6 / Equation 18): After an elaborate description of how the Penman-*
*Monteith approach is used to infer conductances, the simpler (humidity-free) Priestly-*
*Taylor approach is introduced for the Budyko analysis, although alternatives consistent*
*with Penman-Monteith (e.g. FAO grass reference ET) exist. Was there a particular*
*reason for this decision? Luckily it probably affects all sites similarly (more similar than*
*in a global study mixing very humid and arid sites) and seems only to be needed in*
*Fig. 10, even there only slightly changing X axis position but not the overall pattern.*

**7. AR:** We compared the results from the different equations for potential evaporation/ET and we
found differences in annual sums of up to 100 mm $a^{-1}$. This seems to be too uncertain and we
decided to follow the initial idea of Budyko, who assumed that a sufficient amount of available
energy is needed to evaporate precipitation. Therefore, we plotted the relationship between ET/P vs.
$R_N/LP$, with $R_N$ the the annual sum of net radiation. We also added a second figure to the existing
one, given only the arrows of mean ET/P and Rn/P for the two land-uses (AF and MC), the two
methods (ECEB and EC-LC), and the two years (2016 and 2017). Please find the new figure below:

[Figure]

**Figure 11. (a)** Evapotranspiration index (ET/P) versus the radiative dryness index ($R_n/\lambda P$) for both land-uses (AF: filled triangles and dots; MC: empty triangles and dots), both set-ups (ECEB: dots; EC-LC: triangles) and both years (2016: red; 2017: blue). The bold black line describe regions of an energy limitation ($R_n/\lambda P<1$) and a water limitation ($R_n/\lambda P>1$). The arrows indicate mean trends of ET for the effect of different years (black arrow), different methods (blue arrow) and different land-uses (grey arrow). **(b)** Trends of the mean evapotranspiration index (ET/P) versus the mean radiative dryness index ($R_n/\lambda P$) for the effect of different years (black), different methods (blue) and different land-uses (grey) extracted from figure (a).

**8. RC:** Further comments on technical / presentation details:
7. p01L14 (abstract): Consider rewording "superior performance" to make clear that this indicates superior agreement with the widespread EC method. This is not necessarily identical to superior performance in capturing true ET.

**8. AR:** We changed this accordingly:

"Root mean square errors of LE_{EC-LC} vs. LE_{EC} were half as small as LE_{ECEB} vs. LE_{EC}, indicating a superior agreement of the EC-LC set-up with the EC set-up compared to the ECEB set-up."

**9. RC:** *8. p01L17 (abstract): There is an ongoing debate whether, how much and how we should continue to base conclusions about differences on significance (e.g. Amrhein et al., Nature 567:305, DOI: 10.1038/d41586-019-00857-9). While reporting p-values in a paper for the sake of completeness cannot do much harm (without wrong interpretation), care should be taken especially where wrong use in the past was particularly popular, and one of these cases is inferring that a difference is nonexistent or unimportant from a "failed" (insignificant) test. This sentence (and versions of it in the main text) comes somewhat close to suggesting something like this (although not explicitly claiming it). It might be more convincing to show (as done in the main text) how small the difference actually was (maybe compared e.g. to the mean ETs or to the inter-site, inter-period, or inter-method variability that was probably at the bottom of the significance test) and then it could still be mentiond if wanted (here or elsewhere) that the difference was also statistically insignificant (which depends strongly on sample size even if all the means and variances stay the same, and unlike conclusions from significant results, conclusions from insignificant results have the property to become the*

*more likely the smaller the sample size). Also note that if keeping reporting the p-values somewhere, they should be rounded to a reasonable number of digits; especially for the second one at L23, p = 0.0007 or p < 0.001 would be sufficient.*

**9. AR:** We removed the p-values and statistics in the Abstract as shown below:

"[...] With respect to the annual sums of ET over AF and MC, we observed small differences between the two land-uses. We interpret this as an effect of compensating small-scale differences in ET next to and in between the tree strips for ET measurements on system-scale. Most likely, differences in ET rates next to and in between the tree strips are of the same order of magnitude but of opposite sign and compensate each other throughout the year. Differences between annual sums of ET from the two methods were of the same order of magnitude as differences between the two land-uses. Compared to the effect of land-use and different methods on ET, we found larger mean evapotranspiration indices ($\sum$ET/$\sum$P) across sites for a drier than normal year (2016) compared to a wet year (2017). This indicates that we were able to detect differences in ET due to different ambient conditions with the applied methods, rather than the potentially small effect of AF on ET.

We conclude that agroforestry has not resulted in an increased water loss to the atmosphere indicating that agroforestry in Germany can be a land-use alternative to monoculture agriculture without trees. "

**10. RC:** *9. p02L08 (Sect. 1): "comparable" reads strange in this context. Basically they are, aren't they? As far as I know the term monoculture does not distonguish between agriculture and forestry. Also see comment 1.*

**10. AR:** Yes, we removed the term "comparable" and rewrote the sentence:

"SRC plantations are monoculture systems with a single tree species grown."

**11. RC:** *10. p02L32 (Sect. 1): "such as" reads strange in this context. Maybe ", i.e." instead?*

**11. AR:** We changed this accordingly:

"For agroforestry systems we formulated the same hypothesis, i.e. system-scale evapotranspiration over agroforestry systems is higher compared to monoculture agriculture without trees."

**12. RC:** *11. p03L01 (Sect. 1): depend\*s\**

**12. AR:** We changed this accordingly.

**13. RC:** *12. p03L29 (Sect. 2.1): While reporting the access date of an URL is important if that URL is a source of data/information that couldn't be replaced by a better source, in this case the URL more has the role of an advertisement or a reference to further information for interested readers, and what exactly they will find at the project site if it still exists is not relevant to the paper. For this an access date seems inappropriate. If you weren't asked to add it during the access review, I'd suggest to remove it.*

**13. AR:** We were asked to include the access date to all the URL's in a previous publication, so we just added it here as well. It seems to be a journal requirement. But, we can change this later, if required.

**14. RC:** *13. p04L01 (Figure 1): Maybe add a scale bar to the aerial views (or one scale information for all if they are the same). I wonder how wide the elongated MC strip at Forst (b) was, how different the management west and east of it was, and how this is reflected in Sect. 3.2 (footprint analysis).*

**14. AR:** We added a scale at each of the sites because they are not the same. The strip at Forst MC is 48 m wide and the management at the east and the west of this strip was always the same, but different from the MC strip. The crop type at the MC strip was always the same as in between the tree strips at the agroforestry system. As shown in Fig. 3 the footprint extended beyond the MC strip, hence, fluxes at the MC were also affected by the nearby crop fields.

**15. RC:** *14. p05L1 (Table 1): System size. Specify if it refers to AF, MC or the sum of both.*

**15. AR:** The system size referred to the AF system only and we changed it from "System size" to "Agroforestry system size".

**16. RC:** *15. p05L18 (Sect. 2.2.1): Did I understand correctly that this required at least two available Li-7200? If yes clarify, if no reword sentence.*

**16. AR:** Yes, we deployed two LI7200 in parallel in 2017. We clarified this:

"During the field campaigns the standard set-up was extended by an enclosed-path infrared gas analyser (LI-7200 , LI-COR Inc., Lincoln, Nebraska, USA).  In 2016, the campaigns were conducted separately at the AF and MC systems with one available gas analyser, whilst in 2017 both systems were sampled simultaneously with two available gas analyser."

**17. RC:** *16. p06L01 (Table 2): ppp for pressure seems unusually long/complicated. Also, in the row that is solely about ppp it looks a bit lonely (and hard to understand) without the long explanation "Atmospheric pressure". I acknowledge that you aimed at consistently giving the long name only upon first occurrence in the table, but here would be space and reason enough for an exception. Or maybe the row could be switched with the BME280.*

**17. AR:** We named it $P_A$ for atmospheric pressure.

**18. RC:** *17. p07L19: "unpublished data" and then no matching entry in the reference list is a bit vague. If there is not even an internal report to refer to (which could then be listed in the references), "pers. comm." would probably be more appropriate, and at any rate in this case the institutional affiliation of Schmidt et al. should be given e.g. in the acknowledgement, to ensure traceability.*

**18. AR:** We changed this in the main text and in Table A2:

"Marcus Schmidt (pers. comm., Georg August University of Goettingen, Buesgen Institute, Soil Science of Tropical and Subtropical Ecosystems)"

**Table A2.** Site specific soil characteristics, with the soil texture being representative for the top soil column of 0.3 m. The bulk density is representative for the top soil column of 0.05 m. Data provided by Göbel et al. (2018) and Marcus Schmidt (pers. comm., Georg August University of Goettingen, Buesgen Institute, Soil Science of Tropical and Subtropical Ecosystems).

| Site | Clay content (%) | Sand content (%) | Bulk density ($kg\,m^{-3}$) |
|---|---|---|---|
| Dornburg AF | 20.5 | 3.75 | 1.22 |
| Dornburg MC | 38 | 10.75 | 1.19 |
| Forst AF | 7 | 60.75 | 1.3 |
| Forst MC | 9.5 | 66.75 | 1.28 |
| Mariensee AF | 11.75 | 48 | – |
| Mariensee MC | 31.67 | 54.33 | 1.28 |
| Reiffenhausen AF | 23.75 | 31.5 | 1.28 |
| Reiffenhausen MC | 22.75 | 49.75 | 1.28 |
| Wendhausen AF | 35 | 18.25 | 1.085 |
| Wendhausen MC | 44.5 | 27 | 0.89 |

**19. RC:** *18. p07L28 (Sect. 2.2.3): Even though referring to a publication about the method where all this can probably be read in detail, not mentioning that there was a (probably large) spectral loss correction falls back behind the information given in the introduction (p03L17), and will make readers looking for this information in the methods section (the most logical place) wonder if and how this method could work at all.*

**19. AR:** We included more back-ground information on how the set-up worked in this section and for reproducibility also equations on how we transformed RH, TA and PA readings into a water vapour mole fraction into the Appendix.

Please find here the extended section 2.2.3:

**2.2.3 Low-cost eddy covariance (EC-LC)**

The EC-LC set-ups comprised of the same ultrasonic anemometer uSONIC3-omni as used for the EC and ECEB set-ups plus a compact low-cost relative humidity, air temperature and pressure sensor (BME280, BOSCH, Germany, Table 2). Water vapour mole fraction was calculated using measurements of relative humidity, air temperature and air pressure from the low-cost thermohygrometer. A derivation of the water vapour mole fraction from the low-cost thermohygrometer is given in Section A2. The turbulent water vapour fluxes were calculated as the covariance between the vertical wind velocity and the water vapour mole fraction from EC-LC, as per the principle of the eddy covariance method (Baldocchi, 2014). The cheaper but slower thermohygrometer had inferior spectral response characteristics compared to a gas analyser of fast response. The mean spectral correction factor of the thermohygrometer was 42% larger than for the LI-7200 fast response gas analyser for reference, with a 78% larger mean time constant of the thermohygrometer compared to the LI-7200. The mean time constant of the thermohygrometer and the LI-7200 was $2.8\pm1$ s and $0.6\pm0.3$ s, respectively (Markwitz and Siebicke, 2019). Spectral losses in the high-frequency range of the thermohygrometer were corrected by the fully analytical correction method of Moncrieff et al. (1997), which was explicitly recommended for either open-path sensors or closed-path sensors of heated and very short sampling lines. A detailed description and application of the EC-LC set-up for evapotranspiration measurements over AF and MC is given in Markwitz and Siebicke (2019). Evapotranspiration from EC-LC was neither gap-filled for the methodological comparison nor for the analysis of the energy balance closure due to the risk for new errors and artefacts from the respective gap-filling method.

And please find here the Appendix from the revised version of the manuscript:

**A2 Water vapour mole fraction $C_{H_2O_v}$ from the thermohygrometer**

The derivation of the water vapour mole fraction $C_{H_2O_v}$ from relative humidity, air temperature and air pressure from the low-cost thermohygrometer was also presented in Markwitz

and Siebicke (2019) and for completeness given in this section.

The water vapour mole fraction, $C_{H_2O_v}$, was derived from the definition of the specific humidity, q, as the quantity of water vapour per quantity of moist air. The latter two quantities were expressed as the density of water vapour, $\rho_{H_2O_v}$, and moist air, $\rho_m$, respectively. The density of moist air is defined as the sum of the density of dry air, $\rho_d$, and the density of water vapour.

$$q = \frac{\rho_{H_2O_v}}{\rho_m}$$
$$= \frac{\rho_{H_2O_v}}{\rho_d + \rho_{H_2O_v}} \tag{A4}$$

We then replaced the density of water vapour and the density of dry air in Eq. (A4) as per Eqs. (A5) and (A6), respectively,

$$\rho_{H_2O_v} = \frac{C_{H_2O_v} \cdot M_{H_2O_v}}{V_m} \tag{A5}$$

$$\rho_d = \frac{p - e_a}{R_d \cdot T} \tag{A6}$$

with the molar mass of water vapour, $M_{H_2O_v} = 18.02\,\text{g mol}^{-1}$, the molar volume of air

$$V_m = \frac{\Re \cdot T}{p} \ (\text{m}^3\,\text{mol}^{-1}), \tag{A7}$$

the universal gas constant, $\Re = 8.314\,\text{J mol}^{-1}\text{K}^{-1}$, and the specific gas constant of dry air, $R_d = 287.058\,\text{J kg}^{-1}\,\text{K}^{-1}$.

Solving Eq. (A4) for $C_{H_2O_v}$ leads to the water vapour mole fraction

$$C_{H_2O_v} = \frac{q\,\Re\,(p - e_a)}{p\,M_{H_2O_v}\,R_d(1 - q)}. \tag{A8}$$

The specific humidity in Eq. (A8) was calculated as a function of relative humidity, temperature and air pressure measurements from the thermohygrometer:

$$q = 0.622 \cdot \frac{e_a}{p} \tag{A9}$$

The actual vapour pressure, $e_a$ (kPa), in Eq. (A9) was calculated from an approximation of the saturation vapour pressure, $e_*(T)$ (Stull, 1989) and from relative humidity, RH,

$$e = \frac{RH \cdot e_*(T)}{100} \tag{A10}$$

$$e_*(T) = 0.6112 \exp((17.67T)/((T + 273.15) - 29.66)) \tag{A11}$$

$$\tag{A12}$$

**20. RC:** *p08L27 (Sect. 2.3.2): This sentence at a first glance seems to contradict the sentence at the top of the same page. Maybe start like this: "Unlike for the methodological comparison and energy balance analysis, a gap-filling of EC-LC could not be avoided for [this and that, surely not for annual ET sums]. Therefore, for these analyses..."*

**20. AR:** We changed this in the revised version of the manuscript:

> Unlike for the methodological comparison and energy balance analysis, a gap-filling of $ET_{EC-LC}$ could not be avoided for the calculation of annual sums of ET. Therefore, for these analyses we gap-filled the half-hourly $ET_{EC-LC}$ with half-hourly $ET_{ECEB}$ and corrected both $ET_{EC-LC}$ and $ET_{ECEB}$ for the surface energy balance closure as follows

**21. RC:** *p09L10-15 (Sect. 2.4): At the beginning consider replacing "As the" by "By". Citing software tools / packages can be useful when a) advertising that the own code can be made available to the reader or when b) Reproduction of results depends on using the same tool (mention package, e.g. because the method is so complicated it might give different results in other langauges). The major axis however is a statistical term independent of and introduced before R, and if correctly implemented in the package should yield the same result as any self-written implementation. Therefore it seems more important to refer to a statistical textbook or paper - e.g. Webster 1997, Eur. J. Soil Sci. 48:557, doi:10.1111/j.1365-2389.1997.tb00222.x (which by the way also provides in its "calibration" section support for your decision in other places to treat variables to be filled as "dependent" (Y) variables in a regression).*

**21. AR:** We considered the publication and changed the text:

> **2.4 Energy balance closure estimation**
>
> The energy balance closure (EBC) was quantified in two ways:
>
> 1. As the linear regression between the available energy ($R_N$ - G - S), and the sum of the turbulent flux components (LE + H). We applied the major axis linear regression (Webster, 1997), which assumes equally distributed errors in both time series. We interpret the slope between the available energy and the sum of the turbulent fluxes as the closure of the surface energy balance. A slope of one and an intercept of zero corresponds to perfect energy balance closure. In the present study both the slope and the intercept were considered as variable.

**22. RC:** *p13L01 (Fig. 3): Cannot see MC footprint in subpanel d, is this somehow related to the inavailability of a campaign at Reiffenhausen mentioned at p08L24? But footprint modelling only relies on data measured anyway by the EC-H setup needed for ECEB and EC-LC? Maybe it would be good to state in a prominent place (or each time a particular result seems to be missing, e.g. in Fig. 3d, Fig. 4 between g and h, Table 3 row Mariensee EC-LC, Fig. 5, Table 4, Fig. 9) what was the reason (in most cases it seems to be the missing campaign at Reifenhausen MC, but not so e.g. in table 3).*

**22. AR:** The footprint at Reiffenhausen MC is missing due to the unavailability of a campaign there. Yes, the footprint estimation depends only on mentioned variables, but since the campaign did not take place, we decided to not include the data for the particular site and time period. It would distract from the interpretation of ET during the campaign over the AF in Reiffenhausen. In the footprint climatology for the whole year (Fig. A3 in the appendix) we did include Reiffenhausen MC, as this information is used to explain potential differences in annual sums of ET. We clarified where and why data were missing in the following figures and tables of the revised version of the manuscript: Tables 3-5; Figures 3-6, 8, 9, A1, A2, A5

**23. RC:** *p17L26 (Sect. 3.4.2 / describing Fig. 6): Is "square" a commonly recognized or self-explaining description of this kind of diel curve?*

**23. AR:** we rewrote the sentence to:

"The diel cycle of the EBR for Dornburg (Fig. \ref{fig:EnergyBalanceRatio}) show a strong increase between 6 am and 8 am, followed by a positive slope between 8 am and 2 pm, and a strong increase thereafter until 6 pm. The EBR is minimum 0 at 6 am and maximum 1.8 at 6 pm."

**24. RC:** *p20L29 (Sect. 3.4.3): It took (me) several readings to understand how and why you changed magnitudes, after talking about measured data all the time. Basically the idea of this whole section is simpler and more straightforward than it looks, and if needing to shorten the paper, this (writing it simpler or dropping it completely) would be my first suggestion. It can be reduced to the message "the importance of a relative uncertainty in a flux for the EBC scales with the magnitude of that flux". Even this effect probably vanishes when looking at absolute rather than relative errors / uncertainties, and even though it is not completely irrelevant for deciding how much to invest into improving which flux, it could probably also be demonstrated in a more general way with symbolic maths or a thought experiment.*

**24. AR:** We removed this section.

**25. RC:** *p25L04 (Sect. 3.5.2) "related" reads strange in this context, maybe "plots with an ET index".*

**25. AR:** We changed this accordingly:

"The figure indicates first that plots with an ET index larger than one were water limited, [...]"

**26. RC:** *p26L05 (Sect. 3.5.3): "reduce" or "reduced"? The former simply repeats (and takes for granted, but this semms save to me) what the cited references state, while the latter implies a claim that it can be seen well in your data, which should then however be confirmed by a clearer statement.*

**26. AR:** 'reduce' represents better what we wanted to say!

**27. RC:** *p26L01 (Sect. 3.5.3): The methodology section preferred aerodynamic conductance, here aerodynamic resistance (the inverse) is used. Consistently using only resistance or conductance could help to avoid confusion.*

**27. AR:** We changed it to 'Aerodynamic resistance' in the methodology section.

**28. RC:** *p31L06 (acknowledgements): Data from other sites than your own seem to have been used only in one place of the appendix, Fig. A6, if I didn't overlook something. If it is needed at all (there seems to be little connection to the main text), the small amount of sites used there seems to suggest that acknowledgements to the individual site PIs is at least as, or more, important than to the (for this number of sites quite lengthy) list of networks.*

**28. AR:** This was no longer needed, as we removed the figure from the appendix.

**List of major changes in the manuscript**

1. We tidied up the manuscript and strengthened the discussions in the results section.

2. Instead of using any available equation for the calculation of the "potential ET" we decided to follow the initial idea of Budyko, who assumed that a sufficient amount of available energy is needed to evaporate precipitation. Therefore, we plotted the relationship between ET/P and $R_N$/P, with $R_N$ the net radiation, instead of ET/P vs. $ET_P$/P. The different approaches for $ET_P$ gave very different results with differences in $ET_P$ of ~100 mm $a^{-1}$ across the different approaches. We removed related information on the Priestley-Taylor equation from the manuscript.

3. We included a new section in the appendix named derivations. There, we included 1) the derivation of the soil storage flux and the conversion of LE to ET, 2) equations for the calculation of a water vapour mole fraction from relative humidity, air temperature and pressure from the low-cost thermohygrometer and 3) the derivation of the canopy resistance from the rearranged Penman-Monteith equation and the aerodynamic resistance.

4. We removed the descriptive text in the Section on meteorological conditions and phenological state and referred only to the figure with time series and a table with mean values of relevant meteorological parameter from the campaigns.

5. We removed Section 3.4.3 "EBC sensitivity analysis".

6. We changed the letters in the sub-figures to site names, for example Forst AF and Forst MC instead of (a) or (b).

7. We included a new figure (Fig. 8) and a new section 3.5.1 "Sums of evapotranspiration during the campaigns" with cumulative sums of ET from all three methods (EC, EC-LC, ECEB) for the campaign periods at all sites.

8. We included an extra figure (Fig. 7) with diurnal cycle of the energy balance components for Dornburg AF and MC to explain the unexpected diel cycle of the energy balance ratio at the site.

9. We clarified the meaning of the abbreviations AF and MC early in the Abstract and in the introduction, which correspond to short rotation alley cropping agroforestry (AF) and monoculture systems with crops and grasses, but without trees (MC).

[revised manuscript text omitted]